# SEGAUGMENT: Maximizing the Utility of Speech Translation Data with Segmentation-based Augmentations

**Ioannis Tsiamas** and **José A. R. Fonollosa**
Universitat Politècnica de Catalunya, Barcelona
{ioannis.tsiamas,jose.fonollosa}@upc.edu

**Marta R. Costa-jussà**
FAIR Meta, Paris
costajussa@meta.com

## Abstract

End-to-end Speech Translation is hindered by a lack of available data resources. While most of them are based on documents, a sentence-level version is available, which is however single and static, potentially impeding the usefulness of the data. We propose a new data augmentation strategy, SEGAUGMENT, to address this issue by generating multiple alternative sentence-level versions of a dataset. Our method utilizes an Audio Segmentation system, which re-segments the speech of each document with different length constraints, after which we obtain the target text via alignment methods. Experiments demonstrate consistent gains across eight language pairs in MuST-C, with an average increase of 2.5 BLEU points, and up to 5 BLEU for low-resource scenarios in mTEDx. Furthermore, when combined with a strong system, SEGAUGMENT obtains state-of-the-art results in MuST-C. Finally, we show that the proposed method can also successfully augment sentence-level datasets, and that it enables Speech Translation models to close the gap between the manual and automatic segmentation at inference time.

## 1 Introduction

The conventional approach for Speech Translation (ST) involves cascading two separate systems: an Automatic Speech Recognition (ASR) model followed by a Machine Translation (MT) model. However, recent advances in deep learning (Vaswani et al., 2017), coupled with an increased availability of ST corpora (Di Gangi et al., 2019a; Wang et al., 2020a) have enabled the use of *end-to-end* models (Weiss et al., 2017). Although end-to-end models can address several shortcomings of cascaded models, such as slow inference times, error propagation, and information loss, they are limited by a data bottleneck (Sperber and Paulik, 2020). This bottleneck arises from the inability of end-to-end models to directly leverage data from the more

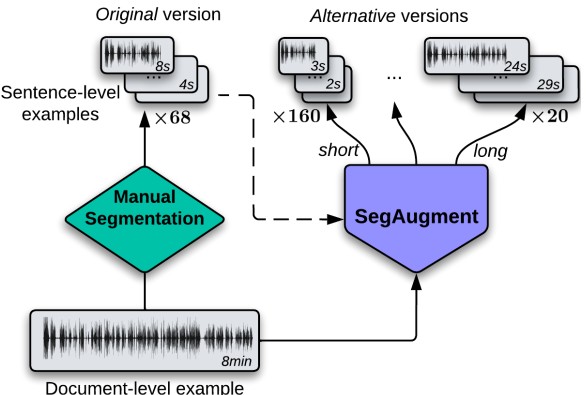

Figure 1: Data Augmentation with SEGAUGMENT.

resourceful tasks of ASR and MT, which restricts them from consistently matching the performance of the cascaded models (Bentivogli et al., 2021; Anastasopoulos et al., 2022, 2021).

The majority of ST corpora are based on document-level speech data, such as MuST-C (Di Gangi et al., 2019a) and mTEDx (Salesky et al., 2021), which are derived from TED talks, with duration times of 10 to 20 minutes. These document-based data are processed into shorter, sentence-level examples through a process called *manual segmentation*, which relies on grammatical features in the text. Still, this sentence-level version is *single* and *static* and is potentially limiting the utility of the already scarce ST datasets.

To address this limitation, we propose SEGAUGMENT, a segmentation-based data augmentation method that generates multiple alternative sentence-level versions of document-level speech data (Fig 1). SEGAUGMENT employs SHAS (Tsiamas et al., 2022b), an Audio Segmentation method that we tune to yield different re-segmentations of a speech document based on duration constraints. For each new segmentation of a document, the corresponding transcript is retrieved via CTC-based forced alignment (Kürzinger et al., 2020), and the target text is obtained with an MT model.

Our contributions are as follows:

- We present SEGAUGMENT, a novel data augmentation method for Speech Translation.
- We demonstrate its effectiveness across eight language pairs in MuST-C, with average gains of 2.5 BLEU points and on three low-resource pairs in mTEDx, with gains up to 5 BLEU.
- When utilized with a strong baseline that combines WAV2VEC 2.0 (Baevski et al., 2020) and MBART50 (Tang et al., 2020), it obtains state-of-the-art results in MuST-C.
- We also show its applicability to data not based on documents, providing an increase of 1.9 BLEU in CoVoST2 (Wang et al., 2021).
- SEGAUGMENT also enables ST models to close the gap between the manual and automatic test set segmentations at inference time.
- Finally, along with our code, we open source all the synthetic data that were created with the proposed method[1].

## 2    Relevant Research

SpecAugment (Park et al., 2019), which directly modifies the speech features by wrapping or masking them, is a standard approach for data augmentation in speech tasks including ST (Bahar et al., 2019; Di Gangi et al., 2019b). WavAugment (Kharitonov et al., 2021) is a similar technique that modifies the speech wave by introducing effects such as pitch, tempo and echo (Gállego et al., 2021). Instead of altering the speech input, our proposed method generates more synthetic data by altering their points of segmentation, and thus is complimentary to techniques such as SpecAugment.

An effective way to address data scarcity in ST is by generating synthetic data from external sources. This can be achieved by using an MT model to translate the transcript of an ASR dataset or a Text-to-Speech (TTS) model to generate speech for the source text of an MT dataset (Jia et al., 2019; Pino et al., 2019; McCarthy et al., 2020). In contrast, SEGAUGMENT generates synthetic data internally, without relying on external datasets.

Previous research has established the benefits of generating synthetic examples by cropping or merging the original ones, with sub-sequence sampling for ASR (Nguyen et al., 2020), and concatenation for MT (Nguyen et al., 2021; Wu et al., 2021; Kondo et al., 2021), as well as for ASR and ST

---

[1]github.com/mt-upc/SegAugment

(Lam et al., 2022a). Our approach, however, segments documents at arbitrary points, thus providing access to a greater number of synthetic examples. An alternative approach by Lam et al. (2022b) involves recombining training data in a linguistically-motivated way, by sampling pivot tokens, retrieving possible continuations from a suffix memory, combining them to obtain new speech-transcription pairs, and finally using an MT model to generate the translations. Our method is similar since it also leverages audio alignments and MT, but instead of mixing speech, it segments at alternative points.

Context-aware ST models have been shown to be robust towards error-prone automatic segmentations of the test set at inference time (Zhang et al., 2021a). Our method bears similarities with Gaido et al. (2020b); Papi et al. (2021) in that it re-segments the train set to create synthetic data. However, unlike their approach, where they split at random words in the transcript, we use a specialized Audio Segmentation method (Tsiamas et al., 2022b) to directly split the audio into segments resembling proper sentences. Furthermore, instead of using word alignment algorithms to get the target text (Dyer et al., 2013), we learn the alignment with an MT model. We thus create high-quality data that can be generally useful, and not only for error-prone test set segmentations. Finally, recent work has demonstrated that context-aware ST models evaluated on fixed-length automatic segmentations can be competitive compared to the manual segmentation (Amrhein and Haddow, 2022). Here, we find that utilizing data from SEGAUGMENT yields high translation quality for ST models evaluated on automatic segmentations, even surpassing the translation quality of the manual segmentation, and without explicitly making them context-aware.

## 3    Background

### 3.1    ST Corpora and Manual Segmentation

A document-level speech translation corpus $\mathcal{D}$ (Di Gangi et al., 2019a; Salesky et al., 2021) is comprised of $n$ triplets that represent the speech wave $\mathbf{X}$, the transcription $\mathbf{Z}$, and the translation $\mathbf{Y}$ of each document.

$$\mathcal{D} = \left\{ (\mathbf{X}^i, \mathbf{Z}^i, \mathbf{Y}^i) \right\}_{i=1}^{n} \qquad (1)$$

In order for the data to be useful for traditional sentence-level ST, the document-level corpus $\mathcal{D}$

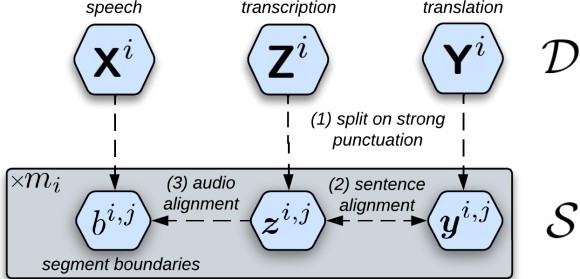

Figure 2: Manual Segmentation of an example $i$ from a document-level corpus $\mathcal{D}$ into $m_i$ examples of a sentence-level corpus $\mathcal{S}$.

is processed to a sentence-level corpus $\mathcal{S}$, with $m = \sum_i^n m_i$ examples.

$$
\begin{aligned}
\mathcal{S} &= \left\{ (\mathbf{x}^i, \mathbf{z}^i, \mathbf{y}^i) \right\}_{i=1}^n \\
&= \left\{ \left\{ (\boldsymbol{x}^{i,j}, \boldsymbol{z}^{i,j}, \boldsymbol{y}^{i,j}) \right\}_{j=1}^{m_i} \right\}_{i=1}^n
\end{aligned} \tag{2}
$$

Where $\mathbf{x}^i = (\boldsymbol{x}^{i,1}, ..., \boldsymbol{x}^{i,m_i})$ are the sentence-level speech waves for the $i$-th document, $\mathbf{z}^i = (\boldsymbol{z}^{i,1}, ..., \boldsymbol{z}^{i,m_i})$ are the sentence-level transcriptions, and $\mathbf{y}^i = (\boldsymbol{y}^{i,1}, ..., \boldsymbol{y}^{i,m_i})$ are the sentence-level translations. Usually, $\mathcal{S}$ is obtained by a process of *manual segmentation* (Fig. 2), where the document-level transcription and translation are split on strong punctuation, and then aligned with cross-lingual sentence alignment (Braune and Fraser, 2010). Finally, the corresponding sentence-level speech waves are obtain by audio-text alignment (McAuliffe et al., 2017). Since speech is continuous, instead of defining the sentence-level speech wave $\boldsymbol{x}^{i,j}$, it is common to define the start and end points $s^{i,j}, e^{i,j} \in \mathbb{R}$ that correspond to the document speech $\mathbf{X}^i$. Thus, $\mathcal{S}$ can be re-defined as:

$$
\mathcal{S} = \left\{ \left( \mathbf{X}^i, \left\{ (b^{i,j}, \boldsymbol{z}^{i,j}, \boldsymbol{y}^{i,j}) \right\}_{j=1}^{m_i} \right) \right\}_{i=1}^n \tag{3}
$$

Where $\mathbf{b}^i = (b^{i,1}, ..., b^{i,m_i})$ is the segmentation for the $i$-th speech wave, and $b^{i,j} = (s^{i,j}, e^{i,j})$ is a tuple of the segment boundaries for the $j$-th segment, for which $\boldsymbol{x}^{i,j} = \mathbf{X}^i_{s^{i,j}:e^{i,j}}$. Note that oftentimes there is a gap between consecutive segments $(e^{i,j}, s^{i,j+1})$ due to silent periods.

## 3.2 Audio Segmentation and SHAS

In end-to-end ST, Audio Segmentation methods aim to find a segmentation $\mathbf{b}'$ for a speech document $\mathbf{X}'$, without making use of its transcription. They are crucial in real world scenarios, when a

test set segmentation is not available, and an automatic one has to be inferred, as simulated by recent IWSLT evaluations (Anastasopoulos et al., 2021, 2022). They usually rely on acoustic features (pause-based), length criteria (length-based), or a combination of both (hybrid). One such hybrid approach is SHAS (Tsiamas et al., 2022b,a), which uses a supervised classification model $\mathcal{C}$ and a hybrid segmentation algorithm $\mathcal{A}$. The classifier $\mathcal{C}$ is a Transformer encoder (Vaswani et al., 2017) with a frozen WAV2VEC 2.0 (Baevski et al., 2020; Babu et al., 2021) as a backbone. It is trained on the speech documents and segment boundaries of a manually-segmented speech corpus, $\mathcal{S}^{\text{SGM}} = \{ (\mathbf{X}^i, \mathbf{b}^i) \}_{i=1}^n$, by predicting whether an audio frame belongs to any of the manually-segmented examples. At inference time, a sequence of binary probabilities $\mathbf{p}'$ is obtained by applying the classifier $\mathcal{C}$ on $\mathbf{X}'$ (eq. 4). Following, parameterized with $thr$ to control the classification threshold, and $\ell = (min, max)$ to control the length of the resulting segments, $\mathcal{A}$ produces the automatic segmentation $\mathbf{b}'$ according to $\mathbf{p}'$ (eq. 5). There are two possible choices for $\mathcal{A}$. The Divide-and-Conquer (PDAC) approach progressively splits the audio at the point $\kappa$ of lowest probability $p'_\kappa > thr$, until all resulting segments are within $\ell$ (Tsiamas et al., 2022b; Potapczyk and Przybysz, 2020). Alternatively, the Streaming (PSTRM) approach takes streams of length $max$ and splits them at the point $\kappa$ with $p'_\kappa > thr$ between $\ell$ or uses the whole stream if no such point exists (Tsiamas et al., 2022b; Gaido et al., 2021).

$$
\mathbf{p}' = \mathcal{C}(\mathbf{X}') \tag{4}
$$
$$
\mathbf{b}' = \mathcal{A}(\mathbf{p}'; min, max, thr) \tag{5}
$$

## 4 Proposed Methodology

The proposed data augmentation method SEGAUGMENT (Fig. 3) aims to increase the utility of the training data $\mathcal{S}$, by generating synthetic sentence-level corpora $\hat{\mathcal{S}}_\ell$, which are based on alternative segmentations of the speech documents in $\mathcal{D}$ (eq. 1). Whereas the manual segmentation (Fig. 2) relies on grammatical features in the text, here we propose to split on acoustic features present in the audio, by utilizing SHAS (§3.2). For the $i$-th speech document $\mathbf{X}^i$, SEGAUGMENT creates alternative segmentation boundaries $\hat{\mathbf{b}}^i$ with SHAS (§4.1), obtains the corresponding transcriptions $\hat{\mathbf{z}}^i$ via CTC-based forced alignment (§4.2), and finally,

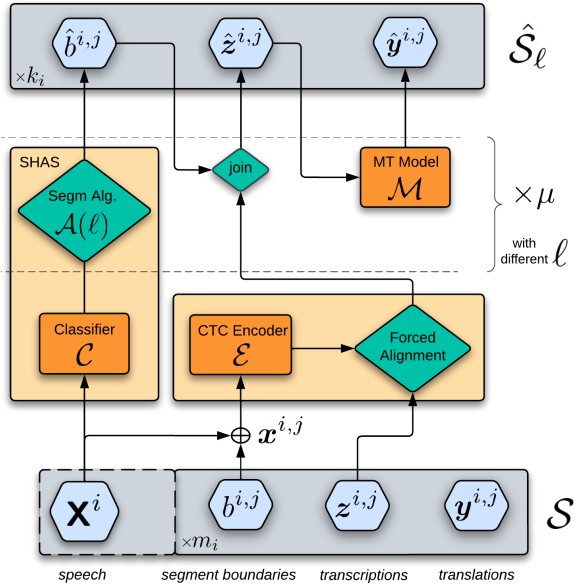

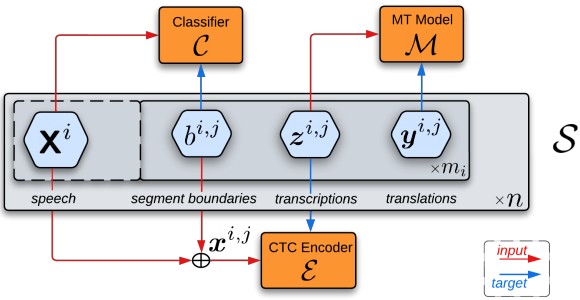

Figure 4: Optional Model Training

Figure 3: The SEGAUGMENT methodology. Given the $i$-th document-level example of $\mathcal{D}$, with $m_i$ sentence-level examples ($\mathcal{S}$), it creates $k_i$ synthetic sentence-level examples by alternative segmentations with SHAS. Changing the segmentation parameters $\ell$ results in several different synthetic corpora $\hat{\mathcal{S}}_\ell$.

generates the translations $\hat{\mathbf{y}}^i$ with an MT model (§4.3). By repeating this process with different parameterizations $\ell$ of the segmentation algorithm $\mathcal{A}$, multiple synthetic sentence-level speech corpora can be generated (§4.4). A synthetic speech corpus $\hat{\mathcal{S}}_\ell$ with $k = \sum_i^n k_i$ examples can be defined as:

$$\hat{\mathcal{S}}_\ell = \left\{ \left( \mathbf{X}^i, \hat{\mathbf{b}}^i, \hat{\mathbf{z}}^i, \hat{\mathbf{y}}^i \right) \right\}_{i=1}^n$$
$$= \left\{ \left( \mathbf{X}^i, \left\{ (\hat{b}^{i,j}, \hat{\mathbf{z}}^{i,j}, \hat{\mathbf{y}}^{i,j}) \right\}_{j=1}^{k_i} \right) \right\}_{i=1}^n \quad (6)$$

Where $\mathbf{X}^i$ is the original speech for the $i$-th document (eq. 1), $\hat{\mathbf{b}}^i = (\hat{b}^{i,1}, ..., \hat{b}^{i,k_i})$ are its alternative segmentations, $\hat{\mathbf{z}}^i = (\hat{\mathbf{z}}^{i,1}, ..., \hat{\mathbf{z}}^{i,k_i})$ are its sentence-level transcriptions, and $\hat{\mathbf{y}}^i = (\hat{\mathbf{y}}^{i,1}, ..., \hat{\mathbf{y}}^{i,k_i})$ are its synthetic sentence-level translations.

In total, three different models are utilized for creating a synthetic corpus, a classifier $\mathcal{C}$ (§3.2) for segmentation, a CTC encoder $\mathcal{E}$ (Graves et al., 2006) for forced alignment, and an MT model $\mathcal{M}$ for text alignment. We can use pre-trained models, or optionally learn them from the manually segmented examples of $\mathcal{S}$ (Fig. 4). The classifier $\mathcal{C}$ can be learned from $\mathcal{S}^{\text{SGM}} = \{(\mathbf{X}^i, \mathbf{b}^i)\}_{i=1}^n$, the encoder $\mathcal{E}$ from $\mathcal{S}^{\text{ASR}} = \{(\mathbf{x}^i, \mathbf{z}^i)\}_{i=1}^n$, and the model $\mathcal{M}$ from $\mathcal{S}^{\text{MT}} = \{(\mathbf{z}^i, \mathbf{y}^i)\}_{i=1}^n$.

Next, we describe in detail the proposed method.

## 4.1 Segmentation

We follow the process described for SHAS (§3.2) and obtain the alternative segmentations $\hat{\mathbf{b}}^i$ for each $\mathbf{X}^i$ in the training corpus, by doing inference with $\mathcal{C}$ and applying the segmentation algorithm $\mathcal{A}$. In contrast to its original use, we use arbitrary values for $min$ to have more control over the length ranges of the segments and prioritize the classification threshold requirements $thr$ over the segment length requirements $\ell$ in the constraint optimization procedure of $\mathcal{A}$, to ensure good data quality.

## 4.2 Audio Alignment

To create the transcriptions $\hat{\mathbf{z}}^i$ of the $i$-th document for the segments $\hat{\mathbf{b}}^i$ (§4.1), we are using CTC-based forced alignment (Kürzinger et al., 2020). We first do inference on the sentence-level speech of the manual segmentation $\mathbf{x}^i = (\boldsymbol{x}^{i,1}, ..., \boldsymbol{x}^{i,n_i})$ with a CTC encoder $\mathcal{E}$, thus obtaining character-level probabilities $\mathbf{u}^i$ for the whole audio. We apply a text cleaning process on the transcriptions $\mathbf{z}^i$, which includes spelling-out numbers, removing unvoiced text such as events and speaker names, removing all remaining characters that are not included in vocabulary, and finally upper casing. The forced alignment algorithm uses the probabilities $\mathbf{u}^i$ and the cleaned text $\mathbf{z}^i$ to find the character segmentation along with the starting and ending timestamps of each entry. Following, we merge characters to words using the special token for the word boundaries, and reverse the cleaning step[2] to recover the original text that corresponds to each segment. For each example $j$, we obtain the source text $\hat{\boldsymbol{z}}^{i,j}$ by joining the corresponding words that are within the segment boundary $\hat{b}^{i,j}$, and apply a post-editing step to fix the casing and punctuation (Alg. 1).

---

[2] We re-introduce any potential noise, e.g. "(Laughter)".

## 4.3 Text Alignment

Unlike the case of manual segmentation (Fig. 2), cross-lingual sentence alignment (Braune and Fraser, 2010) is not applicable, and additionally, word alignment tools (Dyer et al., 2013) yielded sub-optimal results. Thus, we learn the alignment with an MT model $\mathcal{M}$, which is trained on the manually segmented sentence-level data $\mathcal{S}^{\mathrm{MT}} = \{(\mathbf{z}^i, \mathbf{y}^i)\}_{i=1}^n$. The training data is modified by concatenating examples to reflect the length of the examples that will be translated, thus learning model $\mathcal{M}_\ell$ from $\mathcal{S}_\ell^{\mathrm{MT}}$, where $\ell$ are the length parameters used in SHAS. To accurately learn the training set alignment, we use very little regularization, practically overfitting the training data $\mathcal{S}_\ell^{\mathrm{MT}}$ (§A.7). Since there are no sentence-level references available for the synthetic data, we monitor the document-level BLEU (Papineni et al., 2002) in a small number of training documents, and only end the training when it stops increasing. Finally, we obtain the synthetic sentence-level translations $\hat{\mathbf{y}}^i$ with the trained $\mathcal{M}_\ell$.

## 4.4 Multiple Sentence-level Versions

The parameters $\ell = (min, max)$ of the segmentation algorithm $\mathcal{A}$ allow us to have fine-grained control over the length of the produced segments. Different, non-overlapping tuples of $\ell$ result in different segmentations, providing access to multiple synthetic sentence-level versions of each document. Moreover, the additional cost of creating more than one synthetic corpus is relatively low, as the results of the classification with $\mathcal{C}$ and the forced alignment can be cached and reused (Fig. 3).

## 5 Experimental Setup

For our experiments we use data from three ST datasets, MuST-C (Di Gangi et al., 2019a), mTEDx (Salesky et al., 2021) and CoVoST2 (Wang et al., 2021) (Table 1). MuST-C and mTEDx are based on TED talks, and have sentence-level examples, which are derived from document-level ones, via manual segmentation (Fig. 2). CoVoST2 is based on the Common Voice (Ardila et al., 2020) corpus and is inherently a sentence-level corpus. For our main experiments we use eight language pairs from MuST-C v1.0, which are English (En) to German (De), Spanish (Es), French (Fr), Italian (It), Dutch (Nl), Portuguese (Pt), Romanian (Ro), and Russian (Ru). We are also using En-De from v2.0 for certain ablation studies (§A.5, A.6, A.7) and analysis

| Dataset | v | Lang. Pair | # Docs | # Sents | # Hours |
|---|---|---|---|---|---|
| MuST-C | 1.0 | En-De | 2,093 | 234K | 408 |
| | | En-Es | 2,514 | 266K | 496 |
| | | En-Fr | 2,460 | 275K | 485 |
| | | En-It | 2,324 | 254K | 457 |
| | | En-Nl | 2,219 | 248K | 434 |
| | | En-Pt | 2,001 | 206K | 377 |
| | | En-Ro | 2,166 | 236K | 424 |
| | | En-Ru | 2,448 | 265K | 482 |
| | 2.0 | En-De | 2,537 | 251K | 450 |
| mTEDx | | Es-Es | 988 | 102K | 178 |
| | | Pt-Pt | 812 | 90K | 153 |
| | | Es-En | 378 | 36K | 64 |
| | | Pt-En | 279 | 31K | 53 |
| | | Es-Fr | 43 | 4K | 6 |
| CoVoST2 | | En-De | — | 231K | 362 |

Table 1: Training Data Statistics

(§6.8). From mTEDx we use the Es-En, Pt-En, and Es-Fr ST data, as well as the Es-Es, and Pt-Pt for the ASR pre-training, and finally the En-De data from CoVoST2 (Wang et al., 2021).

For segmentation (§4.1) we are using the open-sourced pre-trained English and multilingual SHAS classifiers[3]. For the SHAS algorithm we set the classification threshold $thr = 0.5$ and for length constraints $\ell = (min, max)$, we use four different, non-overlapping tuples of (0.4, 3), (3, 10), (10, 20) and (20, 30) seconds, resulting in *short* (*s*), *medium* (*m*), *long* (*l*), and *extra-long* (*xl*) segmentations. We use PDAC and only apply PSTRM for *xl*, since we observed that PDAC was not able to satisfy the length conditions. For the CTC encoder used in audio alignment (§4.2), we use pre-trained WAV2VEC 2.0 (Baevski et al., 2020) models[4] available on HuggingFace (Wolf et al., 2020). For the MT models used in text alignment (§4.3) we trained *medium*-sized Transformers (Vaswani et al., 2017; Ott et al., 2019) with 6 layers (§A.1.3). When training with SEGAUGMENT, we simply concatenate the four synthetic datasets $\hat{\mathcal{S}}_s$, $\hat{\mathcal{S}}_m$, $\hat{\mathcal{S}}_l$, $\hat{\mathcal{S}}_{xl}$ to the original one ($\mathcal{S}$), and remove any duplicates.

For Speech Translation we train Speech-to-Text Transformer baselines (Wang et al., 2020b). Unless stated otherwise, we use the *small* architecture (s2t_transformer_s) with 12 encoder layers and 6 decoder layers, and dimensionality of 256, with ASR pre-training using only the original data. The full details of the models and the training proce-

---

[3] github.com/mt-upc/SHAS
[4] wav2vec2-large-960h-lv60-self, wav2vec2-large-xlsr-53-spanish, wav2vec2-large-xlsr-53-portuguese

| Model | En-De | En-Es | En-Fr | En-It | En-Nl | En-Pt | En-Ro | En-Ru | Average |
|---|---|---|---|---|---|---|---|---|---|
| Fairseq ST | 22.7 / — | 27.2 / — | 32.9 / — | 22.7 / — | 27.3 / — | 28.1 / — | 21.9 / — | 15.3 / — | 24.8 / — |
| Baseline | 23.2 / 50.5 | 27.5 / 55.0 | 33.1 / 58.3 | 22.9 / 50.4 | 27.3 / 53.9 | 28.7 / 54.9 | 22.2 / 49.0 | 15.3 / 40.4 | 25.0 / 51.6 |
| + SEGAUGMENT | 25.0 / 52.6 | 29.3 / 56.8 | 35.7 / **60.7** | 25.6 / **52.9** | **30.3** / **57.0** | 31.8 / **57.8** | **25.0** / 51.8 | **16.8** / **42.5** | 27.4 / **54.0** |
| + ASR SEGAUGMENT | **25.5** / **52.8** | **29.5** / **56.9** | **35.8** / **60.7** | **25.7** / **52.9** | 30.0 / 56.8 | 31.6 / 57.6 | **25.0** / **52.0** | 16.7 / 42.4 | **27.5** / **54.0** |

Table 2: BLEU($\uparrow$) / chrF2($\uparrow$) scores on MuST-C v1.0 `tst-COMMON`. In **bold** is the best score. All results with SEGAUGMENT and ASR SEGAUGMENT are statistically different from the Baseline with $p < 0.001$. All models use the same architecture and results of Fairseq ST are from Wang et al. (2020b).

dures are available in §A.1.1. For inference, we average the 10 best checkpoints on the validation set and generate with a beam of 5. We evaluate with BLEU[5] (Papineni et al., 2002) and chrF2[6] (Popović, 2017) using SACREBLEU (Post, 2018), and perform statistical significance testing using paired bootstrap resampling (Koehn, 2004) to ensure valid comparisons. To evaluate on an automatic segmentation (§6.5), the hypotheses are aligned to the references of the manual segmentation with MW-ERSEGMENTER (Matusov et al., 2005).

## 6 Results

### 6.1 Main Results in MuST-C

We compare ST models trained with and without SEGAUGMENT on the eight language pairs of MuST-C v1.0, and include results from Wang et al. (2020b), which use the same model architecture. In Table 2, we observe that models leveraging SEGAUGMENT achieve significant and consistent improvements in all language pairs, thus confirming that the proposed method allows us to better utilize the available ST data. More specifically, the improvements range from 1.5 to 3.1 BLEU, with an average gain of 2.4 points. We also investigate the application of SEGAUGMENT during the ASR pre-training, which brings further gains in four language pairs, but the average improvement is only marginal over just using the original ASR data.

### 6.2 Results with SOTA methods

Here we study the impact of the proposed method, when combined with a strong ST model. We use a model with 24 encoder layers, 12 decoder layers, and dimensionality of 1024, where its encoder is initialized from WAV2VEC 2.0 (Baevski et al., 2020) and its decoder from MBART50 (Tang et al., 2020) (§A.1.2). We fine-tune this model end-to-end

| Model | En-Es | En-Fr |
|---|---|---|
| Chimera (Han et al., 2021) | 30.6 / — | 35.6 / — |
| STEMM (Fang et al., 2022) | 31.0 / — | 37.4 / — |
| ConST (Ye et al., 2022) | 32.0 / — | 38.3 / — |
| STPT (Tang et al., 2022) | 33.1 / — | 39.7 / — |
| SpeechUT (Zhang et al., 2022b) | 33.6 / — | 41.4 / — |
| w2v-mBART | 33.3 / 60.2 | 40.8 / 64.5 |
| + SEGAUGMENT | **33.7** / **60.7** | **41.5** / **65.1** |

Table 3: BLEU($\uparrow$) / chrF2($\uparrow$) scores on MuST-C v1.0 `tst-COMMON`. In **bold** is the best score. Results with SEGAUGMENT are statistically different from the w2v-mBART baseline model with $p < 0.001$, apart from the BLEU score for En-Es (33.7) which is with $p < 0.005$.

on ST using MuST-C v1.0, with and without the synthetic data of SEGAUGMENT, and also provide results from other state-of-the-art (SOTA) methods, such as Chimera (Han et al., 2021), STEMM (Fang et al., 2022), ConST (Ye et al., 2022), STPT (Tang et al., 2022), and SpeechUT (Zhang et al., 2022b). As shown in Table 3, despite already having a competitive baseline (w2v-mBART), utilizing SEGAUGMENT achieves further significant improvements[7], reaching SOTA performance in En-Es and En-Fr.

### 6.3 Low-Resource Scenarios

We explore the application of SEGAUGMENT in low-resource and non-English speech settings of mTEDx. In Table 4, we present results of the baseline with and without SEGAUGMENT for Es-En, Pt-En and the extremely low-resource pair of Es-Fr (6 hours). We furthermore provide the BLEU scores from Salesky et al. (2021), which use the *extra-small* model configuration (10M parameters). We use the *extra-small* configuration for Es-Fr, while the others use the *small* one (31M parameters). SEGAUGMENT provides significant improvements in all pairs, with even larger ones when it is also

---

[5]nrefs:1 | bs:1000 | seed:12345 | case:mixed | eff:no | tok:13a | smooth:exp | version:2.3.0

[6]nrefs:1 | bs:1000 | seed:12345 | case:mixed | eff:yes | nc:6 | nw:0 | space:no | version:2.3.0

[7]Compared with §6.1, smaller gains are expected since the effect of data augmentation diminishes with increased data availability, facilitated here by the large pre-trained models.

| Model | Es-En | Pt-En | Es-Fr |
|---|---|---|---|
| Bilingual E2E-ST | 7.0 / — | 8.1 / — | 1.7 / — |
| Baseline | 15.6 / 40.5 | 15.3 / 38.6 | 2.7 / 23.4 |
| + SEGAUGMENT | 18.5 / 43.5 | 17.8 / 41.1 | 3.8 / 23.9 |
| + ASR SEGAUGMENT | **19.1 / 43.9** | **20.3 / 43.7** | **5.3 / 27.2** |

Table 4: BLEU(↑) / chrF2(↑) scores on mTEDx test set. In **bold** is the best score. All results with SEGAUGMENT and ASR SEGAUGMENT are statistically different from the Baseline with $p < 0.001$, with the exception of chrF2 in Es-Fr (23.9) which is with $p < 0.005$. Results of Bilingual E2E-ST are from Salesky et al. (2021).

| Model | En-De |
|---|---|
| Bi-AST (Wang et al., 2021) | 16.3 / — |
| STR (Lam et al., 2022b) | 18.8 / — |
| Baseline | 17.4 / 43.2 |
| + SEGAUGMENT | 19.0 / 44.9 |
| + ASR SEGAUGMENT | **19.3 / 45.4** |

Table 5: BLEU(↑) / chrF2(↑) scores on CoVoST2 test set. In **bold** is the best score. All results with SEGAUGMENT and ASR SEGAUGMENT are statistically different from the Baseline with $p < 0.001$.

utilized during the ASR pre-training, improving the BLEU scores by 2.6-5. Our results here concerning ASR pre-training are more conclusive than in MuST-C (§6.1), possibly due to the better ASR models obtained with SEGAUGMENT (§A.3).

### 6.4 Application on Sentence-level Data

We consider the application of the method to CoVoST, of which the data do not originate from documents. We treat the sentence-level data as "documents" and apply SEGAUGMENT as before. Due to the relatively short duration of the examples, we only apply SEGAUGMENT with *short* and *medium* configurations. In Table 5 we provide our results for En-De, with and without SEGAUGMENT, a bilingual baseline from Wang et al. (2020a), and the recently proposed Sample-Translate-Recombine (STR) augmentation method (Lam et al., 2022b), which uses the same model architecture as our experiments. Although designed for document-level data, SEGAUGMENT brings significant improvements to the baseline[8], even outperforming the STR augmentation method by 0.5 BLEU points.

### 6.5 Automatic Segmentations of the test set

Unlike most research settings, in real-world scenarios, the manual segmentation is not typically available, and ST models must rely on automatic segmentation methods. However, evaluating on automatic segmentation is considered sub-optimal, decreasing BLEU scores by 5-10% (Tsiamas et al., 2022b; Gaido et al., 2021) as compared to evaluating on the manual (gold) segmentation.

Since the synthetic data of SEGAUGMENT originate from an automatic segmentation, we expect they would be useful in bridging the training-inference segmentation mismatch (Papi et al., 2021). We evaluate our baselines with and without SEGAUGMENT on MuST-C tst-COMMON, on both the manual segmentation provided with the dataset, and an automatic one which is obtained by SHAS. In Table 6 we present results with SHAS-*long*, which we found to be the best. Extended results can be found in §A.4. For the purpose of this experiment, we also train another ST model with SEGAUGMENT, where we prepend a special token in each translation, indicating the dataset origin of the example[9]. When generating with such a model, we prompt it with the special token that corresponds to the segmentation of the test set. The results of Table 6 show that the baseline experiences a drop of 1.6 BLEU points (or 6%) on average, when evaluated on the automatic segmentation, confirming previous research (Tsiamas et al., 2022b). Applying SEGAUGMENT, validates our hypothesis, since the average increase of 3.5 BLEU ($23.4 \rightarrow 26.9$) observed in the automatic segmentation is larger than the increase of 2.4 BLEU in the manual one ($25.0 \rightarrow 27.4$). Finally, using SEGAUGMENT with special tokens, enables ST models to reach an average score of 27.3 BLEU points, closing the gap with the manual segmentation (27.4), while being better[10] in three language pairs. To the best of our knowledge, this is the first time[11] that ST models can match (or surpass) the performance of the manual segmentation, demonstrating the usefulness of the proposed method in real-world scenarios. Our results also raise an interesting question, on whether we should continue to consider the manual segmentation as an upper bound of performance for our ST models.

---

[8]There is a potential denoising effect, since CoVoST clips include unvoiced audio either at the start or the end, which our method inherently excludes from the synthetic data.

[9]i.e. <original>, , <m>, <l>, or <xl>

[10]Significance not possible due to different segmentation.

[11]Without context-aware ST (Amrhein and Haddow, 2022).

| Lang. Pair | Model | test set Segmentation | | | |
|---|---|---|---|---|---|
| | | **Manual** | | **SHAS-*long*** | |
| **En-De** | Baseline | 23.2 | | 21.2 | |
| | + SEGAUGMENT | 25.0 | 25.1 | 24.5 | **25.2** |
| | ↪ *special tok.* | 25.1 | | 25.2 | |
| **En-Es** | Baseline | 27.5 | | 26.1 | |
| | + SEGAUGMENT | 29.3 | 29.3 | 29.0 | **29.4** |
| | ↪ *special tok.* | 29.2 | | 29.4 | |
| **En-Fr** | Baseline | 33.1 | | 30.6 | |
| | + SEGAUGMENT | 35.7 | **35.7** | 34.9 | 35.3 |
| | ↪ *special tok.* | 35.6 | | 35.3 | |
| **En-It** | Baseline | 22.9 | | 21.5 | |
| | + SEGAUGMENT | 25.6 | **25.6** | 24.4 | 25.1 |
| | ↪ *special tok.* | 25.0 | | 25.1 | |
| **En-Nl** | Baseline | 27.3 | | 25.7 | |
| | + SEGAUGMENT | 30.3 | **30.3** | 29.7 | 29.8 |
| | ↪ *special tok.* | 29.4 | | 29.8 | |
| **En-Pt** | Baseline | 28.7 | | 26.9 | |
| | + SEGAUGMENT | 31.8 | **31.8** | 31.4 | **31.8** |
| | ↪ *special tok.* | 31.5 | | 31.8 | |
| **En-Ro** | Baseline | 22.2 | | 20.5 | |
| | + SEGAUGMENT | 25.0 | **25.0** | 24.1 | 24.6 |
| | ↪ *special tok.* | 24.8 | | 24.6 | |
| **En-Ru** | Baseline | 15.3 | | 14.6 | |
| | + SEGAUGMENT | 16.8 | 16.8 | 16.8 | **17.0** |
| | ↪ *special tok.* | 16.4 | | 17.0 | |
| **Average** | Baseline | 25.0 | | 23.4 | |
| | + SEGAUGMENT | 27.4 | **27.4** | 26.9 | 27.3 |
| | ↪ *special tok.* | 27.1 | | 27.3 | |

Table 6: BLEU(↑) scores on MuST-C v1.0 `tst-COMMON` with manual and SHAS-*long* segmentation. The second column in Manual and SHAS-*long* is the best score among the three models, and with **bold** is the best score overall for each language pair.

## 6.6 ST without ASR pre-training

Following, we investigate the importance of the ASR pre-training phase, a standard practice (Wang et al., 2020b), which usually is also costly. In Table 7 we present the results of ST models on MuST-C En-Es and En-Fr trained with and without SEGAUGMENT, when skipping the ASR pre-training. We also include the results of the *Revisit-ST* system proposed by Zhang et al. (2022a). We find that models with SEGAUGMENT are competitive even without ASR pre-training, surpassing both the baseline with pre-training and the Revisit-ST system. In general, ASR pre-training could be skipped in favor of using SEGAUGMENT, but including both is the best choice.

| Model | #p | ASR | En-Es | En-Fr |
|---|---|---|---|---|
| Baseline | 31M | ✓ | 27.5 / 55.0 | 33.1 / 58.3 |
| + SEGAUGMENT | 31M | ✓ | 29.3 / 56.8 | 35.7 / 60.7 |
| Revisit-ST | 48M | ✗ | 28.1 / — | 33.4 / — |
| Baseline | 31M | ✗ | 25.3 / 52.3 | 30.1 / 55.4 |
| + SEGAUGMENT | 31M | ✗ | **28.5 / 55.7** | **34.6 / 59.6** |

Table 7: BLEU(↑) / chrF2(↑) scores on MuST-C `tst-COMMON`. In **bold** is the best score among the models without ASR pre-training (✗). Results of Revisit-ST are from Zhang et al. (2022a). #**p** stands for number of parameters.

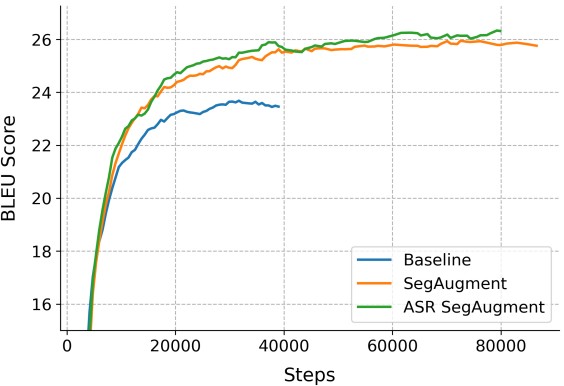

Figure 5: BLEU(↑) scores on MuST-C En-De `dev` during training.

## 6.7 Training Costs

In this section we discuss the computational costs involved with SEGAUGMENT during ST model training. We analyze the performance of models with and without SEGAUGMENT from Table 2, at different training steps in MuST-C En-De `dev`. In Figure 5, we observe that models with our proposed method, not only converge to a better BLEU score, but also consistently surpass the baseline during training. Thus, although utilizing the synthetic data from SEGAUGMENT would naturally result in longer training times, it is still better even when we constraint the available resources.

## 6.8 Analysis

Here we discuss four potential reasons behind the effectiveness of SEGAUGMENT.

**Contextual diversity.** The synthetic examples are based on alternative segmentations and are thus presented within different contextual windows, as compared to the original ones (§A.8). We speculate that this aids the model to generalize more, since phrases and sub-words are seen with less or more context, that might or not be essential for

their translation. Adding additional context that is irrelevant was previously found to be beneficial in low-resource MT, by providing negative context to the attention layers (Nguyen et al., 2021).

**Positional diversity.** With SEGAUGMENT, speech and text units are presented at many more different absolute positions in the speech or target text sequences (§A.11). This is important due to the absolute positional embeddings in the Transformer, which are prone to overfitting (Sinha et al., 2022). We hypothesize that the synthetic data create a diversification effect on the position of each unit, which can be seen as a form of regularization, especially relevant for rare units. This is also supported for the simpler case of example concatenation (Nguyen et al., 2021), while in our case the diversification effect is richer due to the arbitrary document segmentation.

**Length Specialization.** Synthetic datasets created by SEGAUGMENT supply an abundance of examples of extremely long and short lengths, which are relatively infrequent in the original data. This creates a specialization effect enabling ST models trained on the synthetic data to better translate sequences of extreme lengths in the test set (§A.9).

**Knowledge Distillation.** As translations of the synthetic data are generated by MT models, there is an effect similar to that of Knowledge Distillation (KD) (Liu et al., 2019; Gaido et al., 2020a). To quantify this effect, we re-translate the train set of MuST-C En-De four times, with the same MT models employed in SEGAUGMENT. Subsequently, an ST model is trained with the original and re-translated data, referred to as *in-data-KD*, as the MT models did not leverage any external knowledge. In Table 8 we compare in-data-KD with SEGAUGMENT, and find that although in-data-KD provides an increase over the baseline, it exhibits a significant difference of 1 BLEU point with SEGAUGMENT. Our findings confirm the existence of the KD effect, but suggest that SEGAUGMENT is more general as it not only formulates different targets, but also diverse inputs (through re-segmentation), thereby amplifying the positive effects of the source-side contextual and positional diversity. In contrast, KD only provides diversity on the target-side.

## 7 Conclusions

We introduced SEGAUGMENT, a novel data augmentation method that generates synthetic data

| Model | En-De |
|---|---|
| Baseline | 24.3 / 51.9 |
| Baseline + in-data-KD | 25.2 / 52.7 |
| Baseline + SEGAUGMENT | **26.2 / 53.7** |

Table 8: BLEU($\uparrow$) / chrF2($\uparrow$) scores on MuST-C v2.0 En-De `tst-COMMON`. In **bold** is the best score. Results of Baseline + SEGAUGMENT are statistically different from Baseline + in-data-KD with $p < 0.001$.

based on alternative audio segmentations. Through extensive experimentation across multiple datasets, language pairs, and data conditions, we demonstrated the effectiveness of our method, in consistently improving translation quality by 1.5 to 5 BLEU points, and reaching state-of-the-art results when utilized by a strong ST model. Our method was also able to completely close the gap between the automatic and manual segmentations of the test set. Finally, we analyzed the reasons that contribute to our method's improved performance. Future work will investigate the extension of our method to ST for spoken-only languages and Speech-to-Speech translation, by passing the transcription stage.

## Limitations

**General Applicability.** The proposed method requires three steps: Audio Segmentation, CTC-based forced-alignment, and Machine Translation. For Audio Segmentation we used SHAS (Tsiamas et al., 2022b), which requires a classifier that is trained on manually segmented data. Although we demonstrated the method's applicability in CoVoST En-De, which does include a manual segmentation, we used a English classifier that was trained on MuST-C En-De. Therefore, we cannot be certain of the method's effectiveness without manually segmented data for the source language. A possible alternative would be to use a classifier trained on a different source language, since Tsiamas et al. (2022b) showed that SHAS has very high zero-shot capabilities, provided the zero-shot language was also included in the pre-training set of XLS-R (Babu et al., 2021), which serves as a backbone to the classifier. Additionally, we tested our method on several languages pairs, and also on an extremely low-resource one, such as Spanish-French (Es-Fr) in mTEDx, with only 4,000 training examples. Although we showed improvement of

50% in that particular language pair, the two languages involved, Spanish and French, are not by any means considered low-resource. Thus, we cannot be sure about the applicability of the method in truly extremely low-resource languages, such as many African and Native American languages. Furthermore, the current version of the method would not support non-written languages, since the target text is obtained by training a MT model which translated the transcription of each audio segment.

**Biases.** The synthetic data is heavily based on the original data, which may result in inheriting any biases present in the original data. We did not observe any signs of this effect during our research, but we did not conduct a proper investigation to assess the degree at which the synthetic data are biased in any way.

**Computational and Memory costs.** The synthetic data have to be created offline, with a pipeline that involves three different models, resulting in increased computational costs. To reduce these costs, we used pre-trained models for the Audio Segmentation and the CTC encoders, and cached the inference results to be re-used. Thus, the computational cost of creating the synthetic datasets for a given a language pair involves a single inference with the classifier and the CTC encoder, and multiple training/inference phases with MT models. This process can take around 24-36 hours to create four new synthetic datasets for pair in MuST-C, using a single GPU. We acknowledge the computational costs but believe the results justify them. The process could be made much lighter by using non-parametric algorithms in the three steps instead of supervised models, which can be investigated in future work. Finally, despite the computational costs, there is a very small memory cost involved since each synthetic dataset is basically a *txt* file containing the new target text and a *yaml* file containing the new segmentation, only requiring 100-200MB of storage.

## Acknowledgments

Work at UPC was supported by the Spanish State Research Agency (AEI) project PID2019-107579RB-I00 / AEI / 10.13039/501100011033.

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

# A  Appendix

## A.1  Hyperparameters and Training details

### A.1.1  Speech-to-Text Transformer models

For all experiments, apart from §6.2 and from those in mTEDx Es-Fr, we use *s2t_transformer_s* models implemented in FAIRSEQ (Ott et al., 2019; Wang et al., 2020b). They have 12 encoder layers, 6 decoder layers, a dimensionality of 256, a feed-forward dimension of 2048, and 4 heads in the multi-head attention, with 31M parameters in total. We use GELU activations (Hendrycks and Gimpel, 2016) and pre-layernorm (Xiong et al., 2020a). The input is 80-dimensional log-Mel spectrograms, which are processed by a 2-layer convolutional network with 1024 inner channels, output dimension of 256, stride of 2, kernel size of 5, and GLU activations (Dauphin et al., 2017). We do not scale up the output of the convolutional network. The target vocabularies are learned with SentencePiece (Kudo and Richardson, 2018) and have a size of 8,000. We train with AdamW (Loshchilov and Hutter, 2019) with a learning rate of 0.002, a warm-up of 5,000 steps, and an inverse square root scheduler. We use a gradient accumulation of 6, making the effective batch size equal to 320 thousand tokens, apply SpecAugment (Park et al., 2019), and set the dropout to 0.1, applied to attention, activation, and output. The loss function is a standard cross entropy with label smoothing of 0.1. We stop the training when the validation loss does not decrease

for 10 epochs, and average the 10 best checkpoints according to validation BLEU.

The encoders of the ST models are pre-trained on the task of ASR using the same architecture. The only difference is the vocabulary size which is 5,000 and the learning rate which is 0.001. For the models trained without ASR pre-training (§7) we also used a learning rate of 0.001. For MuST-C, we pre-train using the ASR data of En-De, for mTEDx we use the Es-Es and Pt-Pt accordingly, and finally for CoVoST the ASR data of En-De.

For mTEDx Es-Fr we use an *extra-small* architecture (*s2t_transformer_xs*). It has 6 encoder layers and 3 decoder layers, a dimensionality of 256, a feed-forward dimension of 1024, 4 heads in the multi-head attention, and a vocabulary size of 3,000, having a total of 10M parameters. The learning rate is set to 0.001 (warm-up of 500), the batch size to 180 thousand tokens, and use a dropout of 0.2. We also share the weights of the embedding layer and output projection in the decoder. The same model is pre-trained on ASR, but with a dropout of 0.1. All other hyperparameters are the same as for the *small* models described before.

All our experiments were run on a cluster with 8 NVIDIA GeForce rtx 2080 ti. The running times of each experiment on a single GPU ranged from 12 to 36 hours.

### A.1.2   w2v-mBART models

For the experiments of §6.2, we use a strong baseline utilizing pre-trained models and a length adaptor (Tsiamas et al., 2022a). The encoder is composed of a 7-layer convolutional feature extractor and 24-layer Transformer encoder, while the decoder has 12 layers, and a vocabulary of size 250k, with 770M parameters in total. All the layers have an embedding dimensionality of 1024, a feed-forward dimensionality of 4098, GELU activations (Hendrycks and Gimpel, 2016), 16 attention heads, and pre-layer normalization configuration (Xiong et al., 2020b). A strided 1d convolutional layer sub-samples the output of the encoder by 2 times. The encoder is initialized from WAV2VEC 2.0[12] (Baevski et al., 2020), which is pretrained with 60k hours of non-transcribed speech from Libri-Light (Kahn et al., 2020), and fine-tuned for ASR with 960 hours of labeled data from Librispeech (Panayotov et al., 2015). The decoder is initialized from MBART50[13] (Tang et al., 2020), which

---

[12] fairseq/wav2vec/wav2vec2_vox_960h_new.pt
[13] fairseq/models/mbart50/mbart50.ft.1n.tar.gz

---

is fine-tuned En-Xx multilingual machine translation. We fine-tune all the parameters of the model, apart from the feature extractor of the encoder and the embedding layer in the decoder. The inputs to the model are raw waveforms sampled at 16kHz, which are normalized to zero mean and unit variance. We train with AdamW using a base learning rate of 0.0005, with a warm-up for 2,000 steps and an inverse square root scheduler. In the encoder we use 0.1 activation dropout, time masking with probability of 0.2 and channel masking with probability of 0.1 (Baevski et al., 2020). In the decoder we use a dropout of 0.3, and attention dropout of 0.1 (Tang et al., 2020). All other dropouts are not active. The loss function is a standard cross entropy with label smoothing of 0.2. We use gradient accumulation to have an effective batch size of 32M tokens, evaluate every 250 steps, and stop the training when the performance on the validation set does not improve for 20 evaluations. We average the 10 best checkpoints according to the validation BLEU, and generate with a beam search of 5.

### A.1.3   Machine Translation models

For the MT models used for text alignment in SEGAUGMENT (§4.3), we used *medium*-sized Transformers, with 6 encoder and decoder layers, dimensionality of 512, feed-forward dimension of 2048, and 8 heads in the multi-head attention. We train with AdamW using a learning rate of 0.002, with a warm-up for the first 2,500 updates, and the batch size is set to 14 thousand tokens. The vocabulary size is 8,000, and for regularization, we use a small dropout of 0.1 only at the outputs of each layer and label smoothing of 0.1. We stop training when the document-level BLEU does not increase for 20 epochs, average the 10 best checkpoints according to the same metric, and then generate with a beam of size 8.

For the MT models used in the experiments of §A.6, we use a *small* architecture with 6 encoder and decoder layers, with dimensionality of 256, feed-forward dimension of 1024, and 4 heads. We share the weights of the embedding and output projection in the decoder and use a dropout of 0.1 (applied to attention, activation, and output). We stop training when the validation loss does not decrease for 10 epochs, average the 10 best checkpoints according to validation BLEU, and generate with a beam of size 5.

## A.2 CTC-based Forced Alignment Algorithm

Here we provide the algorithm used for audio-text alignment with SEGAUGMENT (§4.2). The complete process for the $i$-th document is summarized in Algorithm 1, where we omit the $i$ index for ease of representation.

---

**Algorithm 1:** CTC-based Forced Alignment

  **input** : $\mathbf{x}$ % List[1d Tensor]
  **input** : $\mathbf{z}$ % List[String]
  **input** : $\hat{\mathbf{b}}$ % List[Tuple[Float, Float]]
  **output** : $\hat{\mathbf{z}}$ % List[String]
1  $\mathbf{u} \leftarrow \mathcal{E}(\mathbf{x})$
2  $\mathbf{z} \leftarrow \text{CLEAN}(\mathbf{z})$
3  $chars \leftarrow \text{FORCED\_ALIGNMENT}(\mathbf{z}, \mathbf{u})$
4  $words \leftarrow \text{JOIN\_CHARS}(chars)$
5  $words \leftarrow \text{REVERSE\_CLEAN}(words)$
6  **for** $j \leftarrow 1$ **to** $len(\hat{\mathbf{b}})$ **do**
7     $\hat{\mathbf{z}}^j \leftarrow \text{JOIN\_WORDS}(\hat{b}^j, words)$
8     $\hat{\mathbf{z}}^j \leftarrow \text{POST\_EDIT}(\hat{\mathbf{z}}^j)$
9  **return** $\hat{\mathbf{z}}$

---

## A.3 Results of ASR pre-training

In Table 9 we present the results of the ASR pre-training, where we observe that by including data from SEGAUGMENT, all ASR models perform better in terms of WER. This indicates that the proposed methods is also useful for ASR augmentations, probably due to the source-side contextual and positional diversity (§6.8).

| Model | MuST-C | mTEDx | | CoVoST |
|---|---|---|---|---|
| | En-De | Es-Es (xs) | Pt-Pt | En-De |
| Baseline | 11.1 | 18 (19.7) | 28.3 | 26.6 |
| + SEGAUGMENT | **10.4** | **16.6 (18.1)** | **23.8** | **23.8** |

Table 9: WER($\downarrow$) scores for ASR models. *xs* indicates the extra-small models pre-trained for mTEDx Es-Fr.

## A.4 Extended Results on Automatic Segmentation of MuST-C

In Table 10 we provide an extended version of Table 6, where we evaluate the models on all four SHAS segmentations of the test set. Regarding the choice of automatic segmentation, we find that the *long* segmentation is the best when using SEGAUGMENT with special tokens (27.3 BLEU), *long* and *extra-long* are equally good without special tokens (26.9 BLEU) and the *medium* segmentation is best without SEGAUGMENT (23.7 BLEU). Using the special tokens reduces the performance on the manual segmentation by 0.3 BLEU (27.4 → 27.1) but

| Lang. Pair | Model | test set Segmentation | | | | | | |
|---|---|---|---|---|---|---|---|---|
| | | Manual | | SHAS | | | | |
| | | — | Best | *s* | *m* | *l* | *xl* | Best |
| **En-De** | Baseline | 23.2 | | 20.6 | 21.7 | 21.2 | 20.2 | |
| | + SEGAUGMENT | 25.0 | 25.1 | 23.1 | 24.3 | 24.5 | 24.7 | **25.2** |
| | ↪ *special tok.* | 25.1 | | 23.1 | 24.5 | 25.2 | 25.1 | |
| **En-Es** | Baseline | 27.5 | | 25.0 | 26.4 | 26.1 | 24.8 | |
| | + SEGAUGMENT | 29.3 | 29.3 | 27.3 | 28.7 | 29.0 | 28.9 | **29.4** |
| | ↪ *special tok.* | 29.2 | | 26.6 | 28.6 | 29.4 | 29.1 | |
| **En-Fr** | Baseline | 33.1 | | 30.0 | 31.0 | 30.6 | 29.0 | |
| | + SEGAUGMENT | 35.7 | **35.7** | 32.3 | 34.1 | 34.9 | 34.8 | 35.3 |
| | ↪ *special tok.* | 35.6 | | 31.4 | 33.8 | 35.3 | 35.1 | |
| **En-It** | Baseline | 22.9 | | 20.0 | 21.6 | 21.5 | 20.3 | |
| | + SEGAUGMENT | 25.6 | **25.6** | 22.7 | 24.2 | 24.4 | 24.8 | 25.1 |
| | ↪ *special tok.* | 25.0 | | 22.3 | 24.3 | 25.1 | 24.5 | |
| **En-Nl** | Baseline | 27.3 | | 24.9 | 26.5 | 25.7 | 24.3 | |
| | + SEGAUGMENT | 30.3 | **30.3** | 27.5 | 29.2 | 29.7 | 29.7 | 29.8 |
| | ↪ *special tok.* | 29.4 | | 26.9 | 26.9 | 29.8 | 29.3 | |
| **En-Pt** | Baseline | 28.7 | | 25.8 | 27.1 | 26.9 | 25.4 | |
| | + SEGAUGMENT | 31.8 | **31.8** | 29.2 | 30.7 | 31.4 | 31.1 | **31.8** |
| | ↪ *special tok.* | 31.5 | | 28.8 | 30.8 | 31.8 | 31.3 | |
| **En-Ro** | Baseline | 22.2 | | 19.9 | 20.8 | 20.5 | 18.7 | |
| | + SEGAUGMENT | 25.0 | **25.0** | 22.5 | 23.4 | 24.1 | 24.3 | 24.6 |
| | ↪ *special tok.* | 24.8 | | 22.6 | 24.1 | 24.6 | 24.0 | |
| **En-Ru** | Baseline | 15.3 | | 13.4 | 14.7 | 14.6 | 13.8 | |
| | + SEGAUGMENT | 16.8 | 16.8 | 15.0 | 16.4 | 16.8 | 16.7 | **17.0** |
| | ↪ *special tok.* | 16.4 | | 14.9 | 16.4 | 17.0 | 16.8 | |
| **Avg** | Baseline | 25.0 | | 22.4 | 23.7 | 23.4 | 22.1 | |
| | + SEGAUGMENT | 27.4 | **27.4** | 24.9 | 26.4 | 26.9 | 26.9 | 27.3 |
| | ↪ *special tok.* | 27.1 | | 24.6 | 26.2 | 27.3 | 26.9 | |

Table 10: BLEU($\uparrow$) scores on manual- and SHAS-segmented MuST-C v1.0 `tst-COMMON`.

increases it in the SHAS-*long* segmentation by 0.4 BLEU (26.9 → 27.3), thus bridging the gap with the manual segmentation.

## A.5 Results with different SEGAUGMENT combinations

We perform ablations by training ST models on different combinations of the original and the synthetic data of SEGAUGMENT, using MuST-C v2.0 En-De. In Table 11, row 1 is essentially the baseline and row 17 is the baseline + SEGAUGMENT. Firstly, we notice that although training single synthetic datasets, but without the original one (rows 2-5) is inferior, the drop is relatively small, considering the absence of the original data. When training with all of them (row 6) actually surpasses the baseline by 1.1 BLEU, without even using the original data. This showcases that SEGAUGMENT is generating high quality data, and even by themselves can produce very good ST models. Secondly, the results show that the more synthetic datasets we use, the better the performance, and that no combination is better than simply using all of them (row 17). We also find that when using 1 or 2 synthetic datasets, the combinations involving longer segmentations (rows 8, 9, 10 and 13, 14) tend to perform better than those involving shorter segmen-

| | Original Data | SEGAUGMENT | | | | BLEU |
|---|---|---|---|---|---|---|
| | | *s* | *m* | *l* | *xl* | |
| (1) | ✓ | | | | | 24.3 |
| (2) | | ✓ | | | | 22.6 |
| (3) | | | ✓ | | | 23.4 |
| (4) | | | | ✓ | | 22.4 |
| (5) | | | | | ✓ | 22.7 |
| (6) | | ✓ | ✓ | ✓ | ✓ | 25.4 |
| (7) | ✓ | ✓ | | | | 24.8 |
| (8) | ✓ | | ✓ | | | 25.1 |
| (9) | ✓ | | | ✓ | | 25.1 |
| (10) | ✓ | | | | ✓ | 25.1 |
| (11) | ✓ | ✓ | ✓ | | | 25.6 |
| (12) | ✓ | | ✓ | ✓ | | 25.6 |
| (13) | ✓ | ✓ | | | ✓ | 25.8 |
| (14) | ✓ | | | ✓ | ✓ | 25.8 |
| (15) | ✓ | ✓ | ✓ | ✓ | | 26.0 |
| (16) | ✓ | | ✓ | ✓ | ✓ | 26.0 |
| (17) | ✓ | ✓ | ✓ | ✓ | ✓ | **26.2** |

Table 11: BLEU($\uparrow$) scores on `tst-COMMON` MuST-C v2.0 En-De for models trained with different combinations of the original and the synthetic data.

| | Original Data | SEGAUGMENT | | | | WER($\downarrow$) | BLEU($\uparrow$) |
|---|---|---|---|---|---|---|---|
| | | *s* | *m* | *l* | *xl* | | |
| (1) | ✓ | | | | | 11.1 | 31.0 |
| (2) | ✓ | ✓ | | | | **10.2** | 31.3 |
| (3) | ✓ | | ✓ | | | 10.3 | 31.6 |
| (4) | ✓ | | | ✓ | | 11.0 | 31.7 |
| (5) | ✓ | | | | ✓ | 10.9 | 31.7 |
| (6) | ✓ | ✓ | ✓ | ✓ | ✓ | 10.4 | **32.1** |

Table 12: WER scores for ASR models and BLEU scores for MT models in MuST-C v2.0 En-De `tst-COMMON`.

tations (rows 7 and 11, 12), while for combinations of 3 synthetic (rows 15, 16), we do not observe any differences.

## A.6 Results on ASR and MT

We investigate the impact of the synthetic data when training ASR and MT models. For this experiments we are using the ASR data and bitext from MuST-C v2.0 En-De. The results of Table 12 show that the synthetic data are in general useful, with a 0.7 reduction in WER and a 1.1 increase in BLEU (row 6). This indicates that the improvements in ST are stemming from both the source side (speech, ASR) and the target side (text, MT). We furthermore notice that longer synthetic examples are better for MT, similar to ST in Table 2, while shorter ones are better for ASR. We hypothesize that this is a consequence of the significance of long context for each task; ASR models tend to use local context (Zhang et al., 2021b), while MT models oftentimes have to use long context, as in pronoun resolution (Voita et al., 2018).

## A.7 Results of MT models used in SEGAUGMENT

In table 13 we present results for the MT models trained to generate the new translation for the alternative data of SEGAUGMENT for MuST-C v2.0 En-De. For each parameterization $\ell = (min, max)$ of the segmentation algorithm $\mathcal{A}$, we train specialized model $\mathcal{M}_\ell$, as described in §4.3. We evaluate on the original training and development sets from $\mathcal{S}$, as well as on the synthetic training set ($\hat{\mathcal{S}}_\ell$), which basically indicates the quality of the target text in the synthetic data. We present both sentence- and document-level BLEU scores for the original data, and only document-level scores for the synthetic ones (since no sentence-level references are available). We notice that MT models obtain very high scores in the original train set, as compared to the development set, indicating that the model has indeed overfitted. In any other case that would be very bad news, but here we willingly overfit the model as our goal is to learn the training set text alignment, and not having a good and generalizable MT model. By looking at the document-level BLEU on the synthetic training set, we can confirm that the MT models have indeed accurately learned the alignments and thus have generated high-quality translation, that can be utilized during ST training.

## A.8 Contextual Windows of SEGAUGMENT Data

We can categorize the new context of the synthetic data of SEGAUGMENT in four types, depending on the type of overlap between the segmentation boundaries $\hat{\mathbf{b}}$ for each document: (1) isolated, when being a subset of an original one, (2) expanded, when being a superset of an original one, (3): mixed, when overlapping with an original one, (4): equal, when being in exactly the same

| MT model $\mathcal{M}_\ell$ | Original train | | Original development | | Synthetic train ($\hat{\mathcal{S}}_\ell$) | |
|---|---|---|---|---|---|---|
| | **BLEU** | **doc-BLEU** | **BLEU** | **doc-BLEU** | **BLEU** | **doc-BLEU** |
| *s* | 95.1 | 79.2 | 24.5 | 26.2 | - | 68.3 |
| *m* | 90.0 | 74.4 | 26.0 | 27.5 | - | 70.4 |
| *l* | 72.9 | 61.3 | 25.5 | 27.0 | - | 64.6 |
| *xl* | 59.6 | 51.8 | 25.9 | 27.4 | - | 56.8 |

Table 13: MT models trained for generating the target text of SEGAUGMENT data. BLEU scores for MuST-C v2.0 En-De training and development set, and synthetic training set. BLEU scores are calculated both at sentence- and document-level.

context as an original one. In Table 14 we apply this categorization for MuST-C v2.0 En-De, and observe that indeed most of the alternative examples form SEGAUGMENT are presented in a new context, with only a small percentage being equal, which we discard when concatenating the datasets for training.

| SEGAUGMENT | **Expanded** | **Isolated** | **Mixed** | **Equal** |
|---|---|---|---|---|
| *s* | 12.3% | 56.9% | 22.3% | 8.5% |
| *m* | 31.1% | 33.3% | 24.3% | 11.3% |
| *l* | 63.2% | 7.9% | 24.6% | 4.2% |
| *xl* | 55.7% | 1.2% | 43.0% | 0.1% |
| | 28.3% | 38.7% | 25.0% | 8.0% |

Table 14: Percentage of examples (rows) for each type of contextual overlap (columns) in the alternative datasets $\hat{\mathcal{S}}_\ell$ compared to the manual one $\mathcal{S}$ for MuST-C v2.0 En-De.

### A.9 Results on duration buckets

In Figure 6 we present evidence for the length specialization effect (§6.8). We construct two *extreme-length* buckets in MuST-C `tst-COMMON`; the first bucket contains all the examples with duration less than 1 second and the second one all those with duration larger than 20 seconds. Then we measure the performance of 5 different types of ST model in the two buckets, which are using the data configurations of rows 1-5 of Table 11. Performance is measured by average BLEU across all eight language pairs of MuST-C. We notice that indeed, models trained with shorter synthetic data ($\hat{\mathcal{S}}_s$, $\hat{\mathcal{S}}_m$) are better at translating the test set bucket with the very short examples, while those trained with longer synthetic data ($\hat{\mathcal{S}}_l$, $\hat{\mathcal{S}}_{xl}$) are better at translating the bucket with the very long ones.

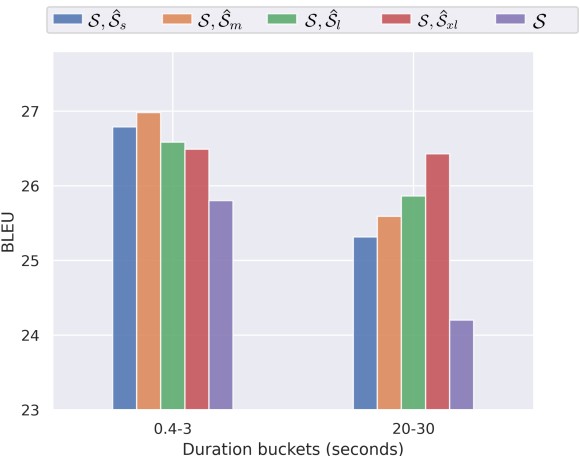

Figure 6: BLEU in two duration buckets of MuST-C `tst-COMMON` of models trained with the four different synthetic datasets and with only the original data. The legend indicates on which data each models is trained on. The left bucket contains all the examples with duration less than 1 second, and the right contains all those with duration longer than 20 seconds. BLEU is the average of all eight language pairs in MuST-C.

### A.10 SEGAUGMENT Data Statistics

In Table 15 we present the number of examples created with SEGAUGMENT for all the language pairs used in this research. We only consider examples longer than 0.4 seconds and shorter than 30 seconds. Differences across the MuST-C v1.0 datasets created with the *short* condition are due to empty segments which were removed. Following, in Table 16 we are presenting the average duration of each example in each created dataset. The average is around 3.8 seconds for *short*-segmented datasets, and around 24 seconds for the *exta-long* ones. In both tables, differences in CoVoST are due to the fact that we prioritize more the length conditions $\ell = (min, max)$ in the segmentation algorithm $\mathcal{A}$ (§4.1), while the other datasets are prioritizing the classification condition $thr$. We did this change in order to make sure that the CoVoST data are segmented in a different way, than in the

| Dataset | v | Lang. Pair | Sentence-level version | | | | | |
|---------|---|-----------|:---------:|:---:|:---:|:---:|:---:|:------:|
| | | | | \textsc{SegAugment} | | | | |
| | | | Original | s | m | l | xl | Concat |
| MuST-C | v1.0 | En-De | 225K | 433K | 253K | 120K | 76K | 882K |
| | | En-Es | 260K | 430K | 253K | 120K | 76K | 879K |
| | | En-Fr | 269K | 421K | 253K | 120K | 76K | 871K |
| | | En-It | 248K | 399K | 253K | 120K | 76K | 848K |
| | | En-Nl | 244K | 433K | 253K | 120K | 76K | 882K |
| | | En-Pt | 201K | 433K | 253K | 120K | 76K | 882K |
| | | En-Ro | 231K | 433K | 253K | 120K | 76K | 882K |
| | | En-Ru | 260K | 419K | 253K | 120K | 76K | 868K |
| | v2.0 | En-De | 249K | 401K | 231K | 109K | 70K | 812K |
| mTEDx | | Es-Es | 101k | 150K | 88K | 43K | 27K | 308K |
| | | Pt-Pt | 90K | 136K | 78K | 37K | 24K | 274K |
| | | Es-En | 36K | 53K | 32K | 15K | 10K | 110K |
| | | Pt-En | 30K | 47K | 27K | 13K | 8K | 94K |
| | | Es-Fr | 3.5K | 5.5K | 3K | 1.5K | 1K | 11K |
| CoVoST2 | | En-De | 231K | 504K | 255K | 0 | 0 | 759K |

Table 15: Number of examples in Sentence-level versions for the manual and \textsc{SegAugment} processes.

manual segmentation. Due to this they might not always resemble proper sentences, but at least they are diverse enough to be useful during training.

| Dataset | v | Lang. Pair | Sentence-level version | | | | |
|---------|---|-----------|:---------:|:---:|:---:|:---:|:---:|
| | | | | \textsc{SegAugment} | | | |
| | | | Original | s | m | l | xl |
| MuST-C | v1.0 | En-De | 6.27 | 3.83 | 6.87 | 15.21 | 23.96 |
| | | En-Es | 6.63 | 3.84 | 6.87 | 15.21 | 23.96 |
| | | En-Fr | 6.28 | 3.84 | 6.87 | 15.21 | 23.96 |
| | | En-It | 6.41 | 3.83 | 6.87 | 15.21 | 23.96 |
| | | En-Nl | 6.23 | 3.84 | 6.87 | 15.21 | 23.96 |
| | | En-Pt | 6.50 | 3.84 | 6.87 | 15.21 | 23.96 |
| | | En-Ro | 6.37 | 3.84 | 6.87 | 15.21 | 23.96 |
| | | En-Ru | 6.47 | 3.83 | 6.87 | 15.21 | 23.96 |
| | v2.0 | En-De | 6.27 | 3.75 | 6.83 | 15.16 | 23.94 |
| mTEDx | | Es-Es | 5.90 | 3.79 | 6.91 | 15.24 | 23.86 |
| | | Pt-Pt | 5.81 | 3.71 | 6.86 | 15.17 | 24.03 |
| | | Es-En | 6.06 | 3.87 | 6.91 | 15.27 | 23.78 |
| | | Pt-En | 5.86 | 3.68 | 6.83 | 15.16 | 24.02 |
| | | Es-Fr | 6.08 | 3.80 | 6.89 | 15.10 | 23.84 |
| CoVoST2 | | En-De | 5.61 | 1.82 | 3.68 | — | — |

Table 16: Average duration (in seconds) of examples in Sentence-level versions for the manual and \textsc{SegAugment} processes.

### A.11 Sub-word Positional Frequency

In Figure 7 we present examples of semi-rare sub-words in the target languages in MuST-C v2.0 En-De training set. We are counting their frequency in terms of absolute position in the examples they appear in. We can observe that when using \textsc{SegAugment}, the positional frequency of this sub-words is much more diverse, covering more space in the possible positions in the target sequence. We hy-

pothesize that this effect could be regularizing the model, aiding in each generalization of this sub-words in different positions (§6.8).

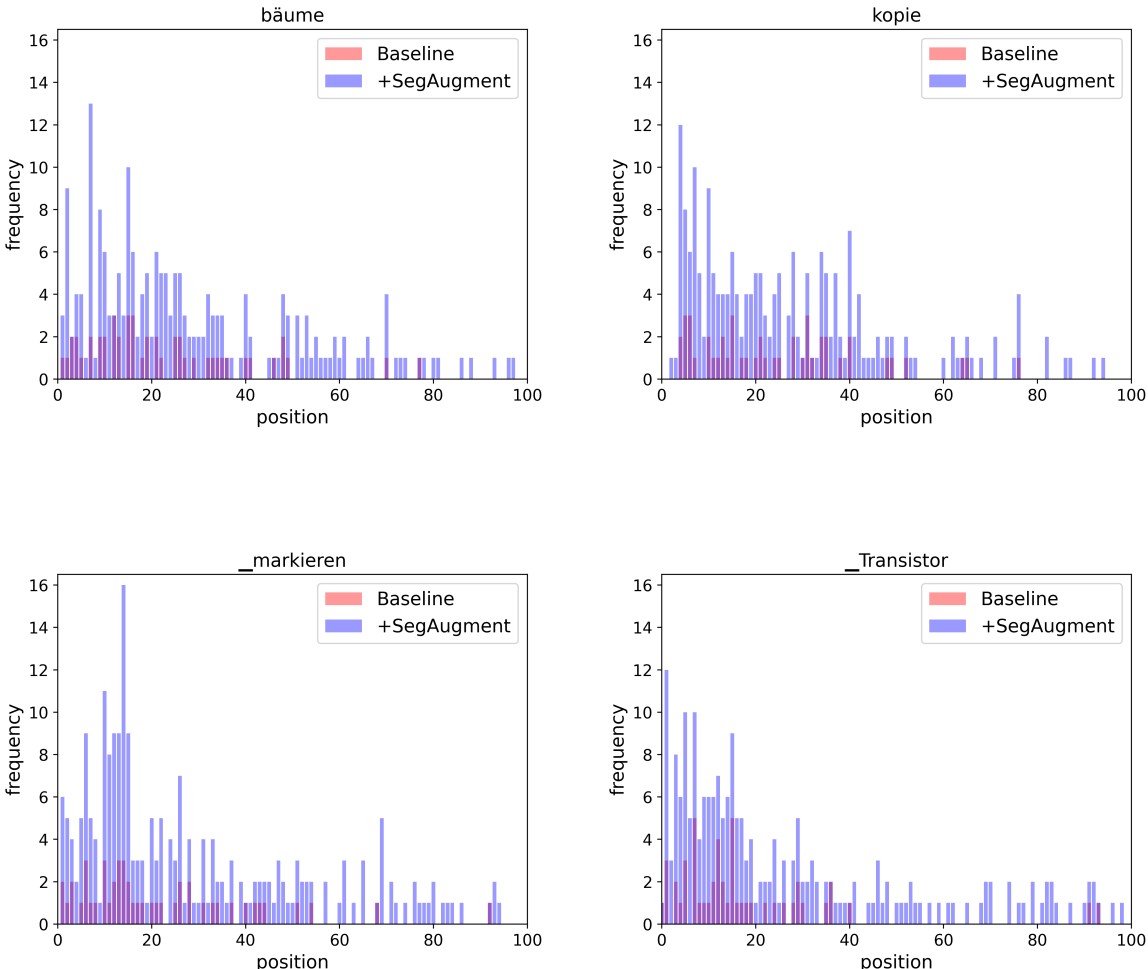

Figure 7: Positional frequency of semi-rare sub-words in the original data with and without SEGAUGMENT in MuST-C v2.0 En-De training set.