# OpenReview forum: "SegAugment: Maximizing the Utility of Speech Translation Data with Segmentation-based Augmentations"
_EMNLP/2023/Conference — EMNLP 2023 Findings_

### Official Review · Reviewer_n8SA · 2023-08-01

**Soundness:** 3

**Excitement:**

3: Ambivalent: It has merits (e.g., it reports state-of-the-art results, the idea is nice), but there are key weaknesses (e.g., it describes incremental work), and it can significantly benefit from another round of revision. However, I won't object to accepting it if my co-reviewers champion it.

**Paper Topic And Main Contributions:**

This work proposes a data augmentation method to alleviate the problem of data scarcity in the speech translation task. When given an arbitrary document-level speech and its manual transcription and translation, the authors use a voice activity detection (VAD) tool to re-segment the whole speech and then obtain sentence-level speeches. For each speech segmentation, they use a speech recognition model to search for the corresponding transcription from the document-level transcription. Then, a machine translation (MT) is used to obtain alignment information between the searched transcription and the tagged document-level translation. They can get the synthetic translation by selecting the document-level BLEU best. Finally, they can change the setting of VAD and repeat the above steps to get multiple speech translation triple data. They carry out the experiments on multiple datasets and languages. The results show that their method can improve upon a strong baseline.

**Questions For The Authors:**

How about comparing or combining other data augmentation methods (e.g., SpecAugment, speed perturb)?

**Reasons To Accept:**

1. The motivation is clear, and the paper is well-organized.

2. The idea is interesting, and the proposed method can benefit the speech translation task.


**Reasons To Reject:**

1. The method has some limitations under several scenarios. For example, this method relies on the MT model, and the translation quality of MT trained with limited data is not good. Thus, the alignment effect will not be ideal enough. Furthermore, if the given speeches are not based on a document or have been shuffled, this method will not work well.

2. The comparison is not sufficient and not fair compared to other methods. The methods compared in Table 2 are weak, and your method is not based on a strong baseline. The model you compared in Table 3 contains about 150M parameters, while your model contains about 800M parameters. Additionally, MBART and w2v-large use much more unlabeled data.


**Reproducibility:**

3: Could reproduce the results with some difficulty. The settings of parameters are underspecified or subjectively determined; the training/evaluation data are not widely available.

**Reviewer Confidence:**

4: Quite sure. I tried to check the important points carefully. It's unlikely, though conceivable, that I missed something that should affect my ratings.

---

> ### Author Rebuttal · Authors · 2023-08-28
>
> Thank you for taking the time to review our work.
>
> We are pleased that you found our paper well-organized and that the motivation behind our work is clear to you. The recognition that our proposed method can potentially benefit the speech translation task is also gratifying.
>
> We genuinely appreciate your feedback and would like to address your concerns and make some clarifications regarding your summary of the paper.
>
> **Clarifications to Reviewer's Summary**
>
> 1. **Segmentation and use of Voice Activity Detection (VAD)**:
>
>     We wish to clarify a misconception regarding segmentation. While traditional Voice Activity Detection (VAD) tools have been used in similar tasks, our study opted for a more sophisticated method. We are using SHAS [1], which leverages self-supervised speech representations from wav2vec 2.0 to craft a specialized classifier combined with a tunable segmentation algorithm. This not only gives us finer control over segment lengths but also ensures accurate segmentations at semantically meaningful points, unlike traditional VADs which have been shown to be error-prone in recent studies [1, 2]
>
> 2. **Generation of Synthetic Translations and use of document-BLEU**:
> We would like to clarify that we do not “get the synthetic translation by selecting the document-level BLEU best”. The best doc-BLEU it's not a tool for generating synthetic translations. Instead, it's employed to monitor the training of our MT alignment model. The absence of references for our new synthetic datapoints led us to rely on their document-level BLEU post-concatenation to determine the best in-training alignment model. More details can be found in section 4.3 “Text Alignment” and Appendix A.7 “Results of MT models used in SEGAUGMENT”.
>
> **Addressing the Reasons to Reject:**
>
> 1. *“The method has some limitations under several scenarios.”*
>
>
>     Although we have discussed extensively the identified limitations of out method in the designated “Limitations” section, the listed limitations here, that are (a) the reliance on MT model for alignment, and (b) effectiveness on sentence-level data, are not valid reasons for concern. We have already provided evidence against them in the paper, but we will elaborate more here.
>
>     1. *“this method relies on the MT model, and the translation quality of MT trained with limited data is not good. Thus, the alignment effect will not be ideal enough.”*
>
>
>         We acknowledge the inherent risks of depending on MT models, especially in low-resource settings. However, our findings from the mTEDx Es-Fr experiments of section 4.3, conducted with a mere 3.5k examples, argue otherwise. The significant increase in translation quality (as evident from Table 4) underscores the robustness of our approach in data-scarce scenarios and consequently indicates the high-quality of the synthetic data and the MT alignment model used to generate them. More concretely, the MT alignment models trained in mTEDx Es-Fr obtain high doc-BLEU scores in the synthetic training set, ranging from 58 to 66 points. Although not directly comparable due to differences in source language and domain, we provide for reference the doc-BLEU scores of the same models for MuST-C En-Fr, which used 270k examples. Despite an 80-fold difference in data size, the scores are not drastically different. The reason that our MT alignment models can still achieve remarkable performance even with limited data is that we deliberately train them past overfitting as we want good alignment within the train set, and not outside of it. Our detailed analysis in section 4.3 “Text Alignment” elaborates on this point. Please also refer to Appendix A.7 “Results of MT models used in SegAugment” for further discussion and analysis on MuST-C En-De.
>
>         *document-BLEU scores on synthetic train set*
>
>         | MT alignment model | mTEDx Es-Fr (3.5k) | MuST-C En-Fr (270k) |
>         | --- | --- | --- |
>         | short | 61.7 | 63.2 |
>         | medium | 66.2 | 71.1 |
>         | long | 58.5 | 68.5 |
>         | extra-long | 58.4 | 61.7 |
>     2. *“if the given speeches are not based on a document or have been shuffled, this method will not work well”*
>
>
>         Our findings in Section 6.4 “Application on Sentence-level Data” offer empirical evidence against this claim and showcase the method's applicability and effectiveness in these scenarios. By using SegAugment in CoVoST En-De, which is a purely sentence-level corpus, we can obtain a significant increase in translation quality (Table 5). Please refer to that section for more details.
>
> 2. *“The comparison is not sufficient and not fair compared to other methods”*
>
>     We understand the importance of establishing the rigor and fairness of our comparisons. We have taken diligent efforts to ensure that our method's comparisons span both "standard" and "state-of-the-art" benchmarks. Our aim has been to transparently demonstrate the efficacy of our method across varying levels of complexities and benchmarks.
>
>     1. *“The methods compared in Table 2 are weak, and your method is not based on a strong baseline.”*
>         - **Objective of Table 2**: The main purpose of showcasing results in Table 2 (section 6.1 “Main Results in MuST-C”) is to demonstrate the benefits of incorporating our proposed method alongside a widely-recognized baseline, namely the fairseq S2T-Transformer [3]. We believe that it's crucial to benchmark our method not just against the state-of-the-art (SOTA) but also against commonly-used, standard baselines. This ensures broader applicability and understanding.
>         - **Regarding the "weak" baseline**: We acknowledge your point on the perceived "weakness" of our baseline. However, our results, particularly with our "weak" baseline augmented with our method, surpass a much more sophisticated model like Chimera [4]. Notably, Chimera utilizes wav2vec 2.0, relies on external MT data, and has about 5 times the training parameters than our implemented baseline here.
>         - **Transitioning to Stronger Baselines**: We recognize the necessity of comparing against cutting-edge techniques. Hence, we present further experiments in section 6.2 “Results with SOTA methods”.
>     2. *“The model you compared in Table 3 contains about 150M parameters, while your model contains about 800M parameters. Additionally, MBART and w2v-large use much more unlabeled data.”*
>         - **Contextualizing Table 3**: The primary message from section 6.2 and the associated Table 3 is the increase in performance that our already strong baseline of w2v-mBART achieves when enhanced with SegAugment. By presenting other top-performing methods in the MuST-C benchmark within the same table, we provide a frame of reference. This allows readers to understand and appreciate the significance of the gains our method offers relative to existing SOTA models.
>         - **Clarifying our Position**: At no point did we imply a direct equivalence between our w2v-mBART model and other SOTA methods listed in Table 3 concerning training parameters or the use of unlabeled data. Our emphasis remains on the tangible improvements our method brings, regardless of the underlying model's complexities.
>
> **Questions for the authors:**
>
> 1. *“How about comparing or combining other data augmentation methods (e.g., SpecAugment, speed perturb)?”*
>
>
>     While traditional data augmentation techniques like SpecAugment [5] focus on directly modifying the speech features of the input, our method focuses on modifying the context and length of an example, both in the input speech and target text. They're complementary, not competing. In fact, SpecAugment has been incorporated into all our models*, highlighting its compatibility.
>
>     *with the exception of w2v-mBART that uses waveforms as input
>
>
> **References**
>
> [1] Tsiamas, Ioannis, Gerard I. Gállego, José A. R. Fonollosa and Marta Ruiz Costa-jussà. “SHAS: Approaching optimal Segmentation for End-to-End Speech Translation.” *Interspeech* (2022).
>
> [2] Marco Gaido, Matteo Negri, Mauro Cettolo, and Marco Turchi. 2021. [Beyond Voice Activity Detection: Hybrid Audio Segmentation for Direct Speech Translation](https://aclanthology.org/2021.icnlsp-1.7). In *Proceedings of the 4th International Conference on Natural Language and Speech Processing (ICNLSP 2021)*, pages 55–62, Trento, Italy. Association for Computational Linguistics.
>
> [3] Changhan Wang, Yun Tang, Xutai Ma, Anne Wu, Dmytro Okhonko, and Juan Pino. 2020. [Fairseq S2T: Fast Speech-to-Text Modeling with Fairseq](https://aclanthology.org/2020.aacl-demo.6). In *Proceedings of the 1st Conference of the Asia-Pacific Chapter of the Association for Computational Linguistics and the 10th International Joint Conference on Natural Language Processing: System Demonstrations*, pages 33–39, Suzhou, China. Association for Computational Linguistics.
>
> [4] Chi Han, Mingxuan Wang, Heng Ji, and Lei Li. 2021. [Learning Shared Semantic Space for Speech-to-Text Translation](https://aclanthology.org/2021.findings-acl.195). In *Findings of the Association for Computational Linguistics: ACL-IJCNLP 2021*, pages 2214–2225, Online. Association for Computational Linguistics.
>
> [5] Park, Daniel S., William Chan, Yu Zhang, Chung-Cheng Chiu, Barret Zoph, Ekin Dogus Cubuk and Quoc V. Le. “SpecAugment: A Simple Data Augmentation Method for Automatic Speech Recognition.” *Interspeech* (2019).

---

### Official Review · Reviewer_bbzS · 2023-08-05

**Soundness:** 4

**Excitement:**

4: Strong: This paper deepens the understanding of some phenomenon or lowers the barriers to an existing research direction.

**Paper Topic And Main Contributions:**

The authors propose a new data augmentation technique for speech translation that generates multiple possible segmentation of the sentences with different length constraints. Methods that are trained on additional data show improvement on two datasets, and show especially improvement on low-resource settings.

**Reasons To Accept:**

- The paper is well-written and easy to follow
- The paper shows strong improvements over baselines and contains much detailed analysis

**Reasons To Reject:**

What is the training time for running the synthetic data with fine-tuning only and with pertaining?
The training time seems to be quite long and with additional pseudo-training data the cost of training seems to be quite high.

**Reproducibility:**

4: Could mostly reproduce the results, but there may be some variation because of sample variance or minor variations in their interpretation of the protocol or method.

**Reviewer Confidence:**

2: Willing to defend my evaluation, but it is fairly likely that I missed some details, didn't understand some central points, or can't be sure about the novelty of the work.

---

> ### Author Rebuttal · Authors · 2023-08-28
>
> We appreciate your recognition of our paper's clarity, improvements over the baselines, and comprehensive analysis. We also understand your concerns regarding the extended training time and the associated costs. Let's address these concerns with specific data:
>
> ### **Impact of SegAugment on Training Time in Speech Translation**
>
> **Note on the experimental setup**: We train all models until convergence, and thus stop the training when the loss on the development set does not improve for 10 consecutive evaluations, where we evaluate every 500 steps.
>
> - **Baseline Model**: Completes training in about 35,000 steps, with a BLEU score of 22.4.
> - **Model with SegAugment**: Completes training around 70,000 steps, with a BLEU score of 24.8.
> - **Model with also ASR-SegAugment**: Completes training at about 80,000 steps, with a BLEU score of 25.5.
>
> Despite almost doubling the resources for the SegAugment models, if we constrain all models to 35k steps, the models with SegAugment still outperform the baseline, achieving BLEU scores of 24.5/24.8. This implies that SegAugment is consistently advantageous, even with resource constraints. The prolonged training duration for models with SegAugment is probably not only due to the larger training set but also due to the challenges of modeling longer examples, with durations ranging from 20-30 seconds.
>
> **BLEU Scores on MuST-C v1.0 En-De dev set (No checkpoint averaging)**:
>
> | model/steps | 10k | 20k | 35k | Final |
> | --- | --- | --- | --- | --- |
> | Baseline | 20.6 | 22.0 | 22.4 | 22.4 |
> | (+) SegAugment | 21.3 | 23.5 | 24.5 | 24.8 |
> | (+) ASR SegAugment | 21.9 | 23.9 | 24.8 | 25.5 |
>
> ### **Impact of SegAugment on Training Time during the ASR Pre-training**
>
> In the ASR pre-training scenario:
>
> - **Baseline**: Takes 45k steps to train.
> - **SegAugment**: Extends to 70k steps.
>
> Although the baseline starts on par with the model that uses SegAugment, by the convergence point (45k steps), the latter one overtakes it. This suggests that while extra resource allocation to train with SegAugment is optional, it ensures superior performance.
>
> **WER Scores on MuST-C v1.0 En-En dev set (No checkpoint averaging)**:
>
> | model/steps | 15k | 30k | 45k | Final |
> | --- | --- | --- | --- | --- |
> | Baseline | 10.6 | 9.0 | 8.6 | 8.6 |
> | (+) SegAugment | 10.6 | 8.7 | 8.1 | 7.6 |
>
> We hope this detailed explanation alleviates your concerns about the impact of SegAugment on the training costs of ST and ASR models. We acknowledge that we did not touch upon this issue in our initial draft and plan to add a section discussing this matter in the final version of the paper or its Appendix.

---

### Official Review · Reviewer_Li15 · 2023-08-10

**Typos Grammar Style And Presentation Improvements:** 1. Relevant Research Section can be b…
**Soundness:** 4

**Excitement:**

4: Strong: This paper deepens the understanding of some phenomenon or lowers the barriers to an existing research direction.

**Paper Topic And Main Contributions:**

Proposing data augmentation technique to generate multiple sentence-level versions of a document-level dataset that vary in length. They show strong results in the MuST-C dataset and robust results in low-resource languages. Then they also talk about sentence-level datasets at the inference time.

Main Contributions:
1. They proposed a data augmentation technique to create varied-length sentence data from document data.
2. Compared the performance in MuST-C and mTEDx datasets.
3. State-of-the-art results in MuST-C
4. Authors also demonstrated effectiveness on sentence-level datasets.
4. Reproducibility with open-source code and dataset.

I will recommend authors to add more information on the relevant research sections. The introduction and Methodology are clear and easy to understand. The authors also clearly explained the limitations of the work but lack any mention of future work.

**Reasons To Accept:**

1. The Methodology is clear and detailed.
2. Strong Results on MuST-C and also on low resource language.
3. Clear Limitations of work.

**Reasons To Reject:**

NA

**Reproducibility:**

4: Could mostly reproduce the results, but there may be some variation because of sample variance or minor variations in their interpretation of the protocol or method.

**Reviewer Confidence:**

4: Quite sure. I tried to check the important points carefully. It's unlikely, though conceivable, that I missed something that should affect my ratings.

---

> ### Author Rebuttal · Authors · 2023-08-28
>
> We would like to express our gratitude for your review and valuable feedback on our work. We are delighted that you found our methodology clear, appreciated the robustness of our results on MuST-C and low-resource languages, and took note of our efforts to maintain transparency with open-source code and dataset availability. We also value the insights you provided regarding the scope for improvement.
>
> In response to your feedback:
>
> 1. **Relevant Research Section**: We acknowledge that the relevant research section is rather compressed right now, since we opted to assign more space to the methodology and experiments. In the camera-ready version, given the additional page availability, we intend to enhance this section with more relevant works. In particular, our extended relevant research section will include more approaches focused on generating synthetic speech [1, 2] and synthetic text [3]. We will also discuss techniques that aim to modify the fbank or waveform features like SpecAugment [4, 5] and WavAugment [6]. Finally, we will touch the usage of MT data with sequence-level knowledge distillation [7], a technique that can be applied in the standard offline way [8, 9] or even online [10].
> 2. **Future Work**: We appreciate your suggestion regarding the elaboration of future directions. In line with this, we aim to expand the discussion on the future research pathways in our revised manuscript.
>     - **Extension to non-written Languages & Speech-to-Speech Translation**: Our ambition is to explore bypassing the transcription stage in the data creation process. Post the segmentation phase with SHAS, we aim to employ an ST model that is overfitted on the training data to generate synthetic translations. This can lift the reliance on the intermediate ASR data, potentially enabling the application of SegAugment on non-written languages.
>     - **Speech-to-Speech Translation**: Given the increased interest in this area [11, 12] and the stark data scarcity, we believe SegAugment's synthetic data can be an asset. We plan to explore its potential for pre-training and even study a potential extension to directly augment speech-to-speech corpora [13, 14]
>     - **Curriculum Learning**: Our preliminary investigations revealed that models trained on “short” synthetic data tend to learn faster. However, their performance tends to plateau or decrease over time compared to models trained on “long” synthetic data. Hence, we plan to explore a curriculum learning [15] strategy that introduces shorter examples initially and gradually transitions to longer examples.
>
> **References**
>
> [1] J. Zhao, G. Haffari, and E. Shareghi. 2023. [Generating Synthetic Speech from SpokenVocab for Speech Translation](https://aclanthology.org/2023.findings-eacl.147). *EACL 2023*, pages 1975–1981, Dubrovnik, Croatia. Association for Computational Linguistics.
>
> [2] A. D. McCarthy, L. Puzon and J. Pino, "SkinAugment: Auto-Encoding Speaker Conversions for Automatic Speech Translation," *ICASSP 2020 - 2020 IEEE International Conference on Acoustics, Speech and Signal Processing (ICASSP)*, Barcelona, Spain, 2020, pp. 7924-7928
>
> [3] C. Mi, L. Xie, and Y. Zhang, "Improving data augmentation for low resource speech-to-text translation with diverse paraphrasing”. Neural Networks, Volume 148, 2022, Pages 194-205, ISSN 0893-6080, https://doi.org/10.1016/j.neunet.2022.01.016.
>
> [4] Park, D.S., Chan, W., Zhang, Y., Chiu, C.-C., Zoph, B., Cubuk, E.D., Le, Q.V. (2019) SpecAugment: A Simple Data Augmentation Method for Automatic Speech Recognition. Proc. Interspeech 2019, 2613-2617, doi: 10.21437/Interspeech.2019-2680
>
> [5] P. Bahar, A. Zeyer, R. Schlüter, and H. Ney, 2019. [On Using SpecAugment for End-to-End Speech Translation](https://aclanthology.org/2019.iwslt-1.22). In *Proceedings of the 16th International Conference on Spoken Language Translation*, Hong Kong. Association for Computational Linguistics.
>
> [6] E. Kharitonov *et al*., "Data Augmenting Contrastive Learning of Speech Representations in the Time Domain," *2021 IEEE Spoken Language Technology Workshop (SLT)*, Shenzhen, China, 2021, pp. 215-222, doi: 10.1109/SLT48900.2021.9383605.
>
> [7] Y. Kim and A. M. Rush. 2016. [Sequence-Level Knowledge Distillation](https://aclanthology.org/D16-1139). In *Proceedings of the 2016 Conference on Empirical Methods in Natural Language Processing*, pages 1317–1327, Austin, Texas. Association for Computational Linguistics.
>
> [8] Liu, Y., Xiong, H., Zhang, J., He, Z., Wu, H., Wang, H., Zong, C. (2019) End-to-End Speech Translation with Knowledge Distillation. Proc. Interspeech 2019, 1128-1132, doi: 10.21437/Interspeech.2019-2582
>
> [9] M. Gaido, M. A. Di Gangi, M. Negri, and M. Turchi. 2020. [End-to-End Speech-Translation with Knowledge Distillation: FBK@IWSLT2020](https://aclanthology.org/2020.iwslt-1.8). In *Proceedings of the 17th International Conference on Spoken Language Translation*, pages 80–88, Online. Association for Computational Linguistics.
>
> [10] Y. Tang, J. Pino, X. Li, C. Wang, and D. Genzel. 2021. [Improving Speech Translation by Understanding and Learning from the Auxiliary Text Translation Task](https://aclanthology.org/2021.acl-long.328). In *Proceedings of the 59th Annual Meeting of the Association for Computational Linguistics and the 11th International Joint Conference on Natural Language Processing (Volume 1: Long Papers)*, pages 4252–4261, Online. Association for Computational Linguistics.
>
> [11] Ann Lee, Peng-Jen Chen, Changhan Wang, Jiatao Gu, Sravya Popuri, Xutai Ma, Adam Polyak, Yossi Adi, Qing He, Yun Tang, Juan Pino, and Wei-Ning Hsu. 2022. [Direct Speech-to-Speech Translation With Discrete Units](https://aclanthology.org/2022.acl-long.235). In *Proceedings of the 60th Annual Meeting of the Association for Computational Linguistics (Volume 1: Long Papers)*, pages 3327–3339, Dublin, Ireland. Association for Computational Linguistics.
>
> [12] Hirofumi Inaguma, Sravya Popuri, Ilia Kulikov, Peng-Jen Chen, Changhan Wang, Yu-An Chung, Yun Tang, Ann Lee, Shinji Watanabe, and Juan Pino. 2023. [UnitY: Two-pass Direct Speech-to-speech Translation with Discrete Units](https://aclanthology.org/2023.acl-long.872). In *Proceedings of the 61st Annual Meeting of the Association for Computational Linguistics (Volume 1: Long Papers)*, pages 15655–15680, Toronto, Canada. Association for Computational Linguistics.
>
> [13] Duquenne, Paul-Ambroise, Hongyu Gong, Ning Dong, Jingfei Du, Ann Lee, Vedanuj Goswani, Changhan Wang, Juan Miguel Pino, Benoît Sagot and Holger Schwenk. “SpeechMatrix: A Large-Scale Mined Corpus of Multilingual Speech-to-Speech Translations.” *Annual Meeting of the Association for Computational Linguistics* (2022).
>
> [14] Ye Jia, Michelle Tadmor Ramanovich, Quan Wang, and Heiga Zen. 2022. [CVSS Corpus and Massively Multilingual Speech-to-Speech Translation](https://aclanthology.org/2022.lrec-1.720). In *Proceedings of the Thirteenth Language Resources and Evaluation Conference*, pages 6691–6703, Marseille, France. European Language Resources Association.
>
> [15] Bengio Y, Louradour J, Collobert R, Weston J (2009) Curriculum learning. In: Proceedings of ICML, pp 41–48

---

### Official Review · Reviewer_uvqE · 2023-08-12

**Soundness:** 5

**Excitement:**

3: Ambivalent: It has merits (e.g., it reports state-of-the-art results, the idea is nice), but there are key weaknesses (e.g., it describes incremental work), and it can significantly benefit from another round of revision. However, I won't object to accepting it if my co-reviewers champion it.

**Paper Topic And Main Contributions:**

This paper presents "SEGAUGMENT", which is a data augmentation method, for speech-to-text translation (S2TT).

"SEGAUGMENT" has 3 steps: 1) segmentation on the document-level speech via "SHAS" algorithm, 2) CTC-based force alignment between the segmented speech and the corresponding transcripts, and 3) translation of the aligned transcripts via a translation model. The output of step 1) and 3) create new speech-translation data pairs for training. This process is repeated by varying the segmentation threshold (e.g., input length) in step1), resulting in data pairs which cover the input-length distribution more thoroughly.

"SEGAUGMENT" is evaluated on three data: 1) MuST-C v1 (a relatively-large corpus with document-level speech), 2) mTEDx (a low-resource corpus with document-level speech) and 3) CoVoST-2 (a relatively-large corpus with sentence-level speech only) in 8, 5, and 1 language direction(s) respectively. In addition to a transformer-based architecture taking log Mel-Fbank as input, "SEGAUGMENT" is also applied on a strong model that contains a "wav2vec2" acoustic encoder and a "mBART50" translation decoder. In all settings, "SEGAUGMENT" obtains strong and statistically significant improvements over competitive baselines.

### Comment
A major weakness of the work is that the procedures of "SEGAUGMENT" are very similar to "Lam et. al 2022: Sample, Translate, [...]" which is also a segment/alignment-based augmentation method for S2TT. A notable difference, algorithmically, is the choice of segmentation point(s) which Lam et. al 2022 focus on the linguistic property of the transcripts, whereas "SEGAUGMENT" is on relatively arbitrary points.

In spite of the similarity, one major contribution of the work is its evaluation and "SEGAUGMENT"'s strong performance on both manually-segmented and automatically-segmented speech, where automatically-segmented speech is a realistic and challenging problem.

Apart from solid improvement over baselines, another strength of the work is its thoroughness in the experimental design. In terms of data, the method is evaluated on diverse language pairs and data sets. In terms of architecture, the method is evaluated on both small architecture (with fbank as input) and large architecture (self-supervised pre-training). Its improvements are also supported by significant tests. There are also interesting side experiments which provide details on the segmentation choices.

**Questions For The Authors:**

- Have you tried replacing "SHAS" in algorithm 1 with other augmentation tools? If yes, what are the results?


**Reasons To Accept:**

- Its evaluation and "SEGAUGMENT"'s strong performance on both manually-segmented and automatically-segmented speech, where automatically-segmented speech is a realistic and challenging problem.
- Its thoroughness in the experimental design, see above.

**Reasons To Reject:**

- "SEGAUGMENT" has procedures that are very similar to "Lam et. al 2022: Sample, Translate, [...]". This may weaken its novelty as a paper about data augmentation.

**Reproducibility:**

4: Could mostly reproduce the results, but there may be some variation because of sample variance or minor variations in their interpretation of the protocol or method.

**Reviewer Confidence:**

4: Quite sure. I tried to check the important points carefully. It's unlikely, though conceivable, that I missed something that should affect my ratings.

**Typos Grammar Style And Presentation Improvements:**

- It might be better to indicate the meaning behind each symbol in the flow-diagrams, e.g. Figure 3.

---

> ### Author Rebuttal · Authors · 2023-08-28
>
> Thank you for the comprehensive review. Your recognition of our method’s strong performance on both manually-segmented and automatically-segmented speech, its thorough experimental design, and the rigorous evaluations across diverse language pairs and datasets is particularly appreciated. Such insights are invaluable for both the paper's refinement and for the broader community's understanding.
>
> **Response to Reasons to Reject**
>
> We understand the concerns regarding the resemblance of SegAugment’s procedures to Sample-Translate-Recombine (STR) [1]. However, some crucial differences and additional strengths are worth highlighting:
>
> 1. **Segmentation Point Difference & Performance on Automatic Segmentations**: Our method focuses on versatile segmentation points as opposed to the linguistic properties centered approach of STR. This fundamental difference not only yields varied training data pairs but also culminates in distinct model behaviors, like contextual & positional diversification and length specialization (Section 6.7 “Analysis”). Most notably, as outlined in section 6.5 "Automatic Segmentations of the test set", models with SegAugment demonstrate high performance gains on automatically segmented test sets for the 8 language pairs of MuST-C. This is a direct consequence of our method’s ability to bridge the training-inference segmentation mismatch, a capability that the STR method does not possess.
> 2. **Complementarity with STR**: While SegAugment and STR share some foundational procedures, their distinct methodologies, and the nature of the synthetic data they create make them complementary in the ST domain, each addressing different challenges. This inherent diversity in approach positions the two methods as complimentary, suggesting a potential combined usage, further improving performance.
> 3. **Thoroughness in Experimental Design**: SegAugment was exhaustively tested across diverse datasets, language pairs, and architectures. This not only validates its versatility but also underscores its novel approach in handling a spectrum of complexities in the field.
>
> **Questions for the Authors & Further Clarifications**
>
> *"Have you tried replacing "SHAS" in algorithm 1 with other augmentation tools? If yes, what are the results?"*
>
> We opted to use SHAS due to its documented efficacy as highlighted in the literature [2, 3]. Furthermore, SHAS provides us the flexibility to control the resulting segment lengths through the min-max parameterarization, which in turn allows us to create multiple synthetic datasets for each training corpus. The combination of these multiple synthetic corpora of different segment lengths is key to the strong performance SegAugment, as highlighted by the ablation study in Appendix A.5 “Results with different SegAugment combinations”.
>
> **Typos Grammar Style And Presentation Improvements**
>
> Your feedback regarding Figure 3 is noted. To ensure clarity, we will revise the figure with detailed annotations for each symbol.
>
> **References**
>
> [1] Tsz Kin Lam, Shigehiko Schamoni, and Stefan Riezler. 2022. [Sample, Translate, Recombine: Leveraging Audio Alignments for Data Augmentation in End-to-end Speech Translation](https://aclanthology.org/2022.acl-short.27). In *Proceedings of the 60th Annual Meeting of the Association for Computational Linguistics (Volume 2: Short Papers)*, pages 245–254, Dublin, Ireland. Association for Computational Linguistics.
>
> [2] Tsiamas, Ioannis, Gerard I. Gállego, José A. R. Fonollosa and Marta Ruiz Costa-jussà. “SHAS: Approaching optimal Segmentation for End-to-End Speech Translation.” *Interspeech* (2022).
>
> [3] Marco Gaido, Sara Papi, Dennis Fucci, Giuseppe Fiameni, Matteo Negri, and Marco Turchi. 2022. [Efficient yet Competitive Speech Translation: FBK@IWSLT2022](https://aclanthology.org/2022.iwslt-1.13). In *Proceedings of the 19th International Conference on Spoken Language Translation (IWSLT 2022)*, pages 177–189, Dublin, Ireland (in-person and online). Association for Computational Linguistics.

---

### Meta-Review · Area_Chair_gZsD · 2023-09-15

**Recommendation:** 4

**Metareview:**

The paper proposes a novel method for data argmentation for speech translation.  By providing different segmentations of the document level triples of the training data, the data size are multiplied.  Intensive experiments are provided showing the effectiveness of the proposed method.  The paper is clearly written and the experiments are well organized and the results show the improvement is significant.   Three reviewers gave positive reviews, while the reviewer n8SA had some concerns on the comparison.  The authors gave detailed response which I think answered the concerns well.

---

### Decision · Program_Chairs · 2023-10-07

**Decision:**

Accept-Findings

**Comment:**

The paper proposes a novel method for data argmentation for speech translation.  By providing different segmentations of the document level triples of the training data, the data size are multiplied.  Intensive experiments are provided showing the effectiveness of the proposed method.  The paper is clearly written and the experiments are well organized and the results show the improvement is significant.   Three reviewers gave positive reviews, while the reviewer n8SA had some concerns on the comparison.  The authors gave detailed response which I think answered the concerns well.